# CADO: From Imitation to Cost Minimization for Heatmap-based Solvers in Combinatorial Optimization

**Hyungseok Song**[*]                                                          *hyungseok.song@lgresearch.ai*
*LG AI Research*

**Deunsol Yoon**[*]                                                                    *dsyoon@lgresearch.ai*
*LG AI Research*

**Kanghoon Lee**                                                            *kanghoon.lee@lgresearch.ai*
*LG AI Research*

**Han-Seul Jeong**                                                          *hanseul.jeong@lgresearch.ai*
*LG AI Research*

**Soonyoung Lee**                                                         *soonyoung.lee@lgresearch.ai*
*LG AI Research*

**Woohyung Lim**[†]                                                                    *w.lim@lgresearch.ai*
*LG AI Research*

**Reviewed on OpenReview:** *https://openreview.net/forum?id=fvxx5FOED6*

## Abstract

Heatmap-based solvers have emerged as a promising paradigm for Combinatorial Optimization (CO). However, we argue that the dominant Supervised Learning (SL) training paradigm suffers from a systematic objective mismatch: minimizing imitation loss (e.g., cross-entropy) does not guarantee solution cost minimization. We dissect this mismatch into two deficiencies: Decoder-Blindness (being oblivious to the non-differentiable decoding process) and Cost-Blindness (prioritizing structural imitation over solution quality). We empirically characterize these intrinsic flaws and show that they impose a performance limitation. To overcome this limitation, we propose CADO (**C**ost-**A**ware **D**iffusion models for **O**ptimization), a streamlined Reinforcement Learning fine-tuning framework that formulates the diffusion denoising process as an MDP to directly optimize the post-decoded solution cost. We introduce Label-Centered Reward, which repurposes ground-truth labels as unbiased baselines rather than imitation targets, and Hybrid Fine-Tuning for parameter-efficient adaptation. CADO achieves state-of-the-art performance across diverse benchmarks, showing that objective alignment significantly improves the quality of SL-trained heatmap solvers.

## 1 Introduction

Combinatorial optimization (CO) problems are notoriously challenging owing to their inherent NP-hardness (Karp, 1975). Neural Combinatorial Optimization (NCO) has emerged as a promising alternative to traditional heuristics (Lin & Kernighan, 1973; Helsgaun, 2017) and exact solvers (Applegate et al., 2006; Vielma, 2015). While **autoregressive solvers**—which iteratively construct solutions by extending partial candidates—have dominated the field with their strong performance (Kool et al., 2019; Kwon et al., 2020; Kim et al., 2022), **heatmap-based solvers** have recently shown significant promise due to their expressive power in modeling

---

[*]Equal contribution.
[†]Corresponding author.

high-dimensional distributions and capability for efficient parallel inference (Joshi et al., 2019; Fu et al., 2021; Geisler et al., 2022). Instead of sequential generation, heatmap-based solvers produce a complete solution probability map (heatmap) in a one-shot manner, which is subsequently discretized into a feasible solution via a lightweight decoding heuristic.

The dominant paradigm for heatmap-based solvers—exemplified by state-of-the-art diffusion models like DIFUSCO (Sun & Yang, 2023)—is **Supervised Learning (SL)**. This approach trains models to generate heatmaps that imitate optimal solutions via surrogate losses (e.g., cross-entropy), treating ground-truth solutions as labels. Although **Reinforcement Learning (RL)** naturally aligns with the true CO objective of cost minimization, SL is generally preferred for its training stability and the implicit hypothesis that heatmaps approximating optimal solutions will be decoded into low-cost solutions.

However, we challenge this premise. While prior works (Xia et al., 2024) noted the mathematical discrepancy between SL and RL objectives, its actual impact on heatmap solver performance has remained unexplored. We empirically characterize this mismatch and show that it leads to performance degradation—closer imitation of optimal solutions does not guarantee lower solution cost. We identify the SL objective's **Decoder-Blindness** (ignoring the decoding heuristic) and **Cost-Blindness** (neglecting the solution cost) as the twin factors behind this degradation.

To resolve this mismatch, we propose an RL fine-tuning framework for SL-based heatmap solvers designed to optimize the post-decoded solution cost, thereby directly overcoming both Decoder-Blindness and Cost-Blindness. We instantiate this framework on diffusion models as CADO (Cost-Aware Diffusion Models for Optimization), which formulates the denoising process as a Markov Decision Process (MDP). Furthermore, we introduce Label-Centered Reward (LCR), which repurposes SL dataset labels as unbiased baselines rather than imitation targets, and Hybrid Fine-Tuning (Hybrid-FT) for stable and efficient RL fine-tuning.

Our contributions are threefold:

- **Empirical Analysis of Objective Mismatch.** We empirically identify and dissect the objective mismatch of SL-based heatmap solvers into **Decoder-Blindness** and **Cost-Blindness**, empirically characterizing this mismatch and showing that it degrades performance.

- **CADO Framework.** We propose CADO, an RL fine-tuning framework for SL-based heatmap solvers that resolves both blindnesses by explicitly minimizing the post-decoded solution cost, thereby aligning the diffusion process with the true CO objective.

- **Stability and State-of-the-Art Performance.** We introduce Label-Centered Reward (LCR) and Hybrid Fine-Tuning (Hybrid-FT) to stabilize RL training. Extensive experiments show that CADO consistently outperforms existing baselines across diverse CO benchmarks.

## 2 Preliminaries

### 2.1 Problem Formulation

A Combinatorial Optimization (CO) problem instance is denoted by $g \in \mathcal{G}$, where $\mathcal{G}$ is the set of all instances. For each instance $g$, the problem is defined over a discrete solution space $\mathcal{X}_g$, typically represented as $\{0, 1\}^{N_g}$. We define the *feasible solution space* $\mathcal{F}_g \subseteq \mathcal{X}_g$ as the set of solutions that satisfy all instance-specific constraints. The task is then to find a solution $\boldsymbol{x} \in \mathcal{F}_g$ that minimizes the cost function $c_g : \mathcal{F}_g \to \mathbb{R}$, defined as:

$$\boldsymbol{x}^{\star} = \arg \min_{\boldsymbol{x} \in \mathcal{F}_g} c_g(\boldsymbol{x}). \tag{1}$$

Since most CO problems are NP-hard, finding the exact optimal solution $\boldsymbol{x}^{\star}$ is computationally intractable. The practical goal is thus to efficiently find a near-optimal solution within a given computational budget. Neural Combinatorial Optimization (NCO) addresses this by learning a (possibly stochastic) policy $\pi_\theta(\boldsymbol{x}|g)$ that models the distribution of feasible solutions $\boldsymbol{x} \in \mathcal{F}_g$ for a given instance $g$. The learning objective is to determine the optimal parameters $\theta^{\star}$ that minimize the expected cost over the distribution of instances:

$$\theta^{\star} = \arg \min_{\theta} \mathbb{E}_{g \sim \mathcal{G}} \left[ \mathbb{E}_{\boldsymbol{x} \sim \pi_\theta(\cdot|g)}[c_g(\boldsymbol{x})] \right]. \tag{2}$$

We describe two specific CO problems as examples: the Traveling Salesman Problem (TSP) and the Maximum Independent Set (MIS) problem. In the TSP, an instance $g$ represents the coordinates of $n$ cities to be visited. A feasible solution $\boldsymbol{x}$ is an $n \times n$ matrix, where $\boldsymbol{x}[i,j] = 1$ if the traveler moves from city $i$ to city $j$ and 0 otherwise. The total solution space is $\mathcal{X}_g = \{0,1\}^{n \times n}$, and the feasible solution space $\mathcal{F}_g$ is the set of all valid TSP tours that visit each city exactly once. The cost function $c_g(\cdot)$ represents the total length of the tour. In the Maximum Independent Set (MIS) problem, an instance $g$ represents a graph $(V_g, E_g)$, where $V_g$ and $E_g$ denote the sets of vertices and edges, respectively. The solution space $\mathcal{X}_g = \{0,1\}^{|V_g|}$ indicates whether each vertex $v \in V_g$ is included in the independent set. To satisfy the independence property, a feasible solution $\boldsymbol{x} \in \mathcal{F}_g$ must not contain any two vertices connected by an edge in $E_g$. To remain consistent with the minimization objective, the cost function $c_g(\cdot)$ is defined as the negative of the total number of selected nodes. Heatmap-based approaches are generally amenable to CO problems where the discrete solution space can be formulated as a fixed-dimensional probability distribution over the graph's components. As illustrated above, for edge-centric problems such as TSP, the solution is represented as an $n \times n$ adjacency matrix, while for node-centric problems such as MIS, it is a 1D binary vector of size $|V_g|$.

## 2.2 Neural Combinatorial Optimization Solver

Neural approaches to CO generally fall into two main categories: autoregressive and heatmap-based solvers. **Autoregressive solvers** construct solutions iteratively by extending partial candidates.

Conversely, **heatmap-based solvers** aim to generate a complete feasible solution $\boldsymbol{x} \in \mathcal{F}_g$ in a single forward pass. Since strictly enforcing hard constraints directly within the network output is challenging, this paradigm employs a two-stage decomposition. First, a **heatmap generator** $\tilde{\pi}_\theta : \mathcal{G} \to [0,1]^{N_g}$ maps an instance $g$ to a probabilistic heatmap $\tilde{\boldsymbol{x}} = \tilde{\pi}_\theta(g)$. Subsequently, a **post-processing decoder** $f_g : [0,1]^{N_g} \to \mathcal{F}_g$ projects $\tilde{\boldsymbol{x}}$ into a discrete solution $\boldsymbol{x} \in \mathcal{F}_g$, typically using a lightweight non-differentiable heuristic. The resulting policy $\pi_\theta(\boldsymbol{x}|g)$ is defined by marginalizing over the heatmap $\tilde{\boldsymbol{x}}$:

$$\pi_\theta(\boldsymbol{x}|g) = \int f_g(\boldsymbol{x}|\tilde{\boldsymbol{x}}) \tilde{\pi}_\theta(\tilde{\boldsymbol{x}}|g) d\tilde{\boldsymbol{x}}. \tag{3}$$

The training procedure optimizes $\tilde{\pi}_\theta$ while keeping the decoder $f_g$ fixed, using either SL or RL objectives.

The **SL objective** leverages a dataset of optimal solutions $\boldsymbol{x}_g^\star$ as labels. Fundamentally, this approach aims to induce the model to generate solutions structurally close to $\boldsymbol{x}_g^\star$. However, the non-differentiable nature of $f_g$ precludes direct likelihood optimization. Instead, models minimize a surrogate objective, training the generator $\tilde{\pi}_\theta$ to approximate the labels (i.e., $\tilde{\boldsymbol{x}} \approx \boldsymbol{x}_g^\star$) via an instance-level loss $\mathcal{L}_{SL}$ (e.g., binary cross-entropy over each heatmap entry):

$$\mathcal{L}(\theta) = \mathbb{E}_{g \sim \mathcal{G}, \tilde{\boldsymbol{x}} \sim \tilde{\pi}_\theta(\cdot|g)} \left[ \mathcal{L}_{SL}(\tilde{\boldsymbol{x}}, \boldsymbol{x}_g^\star) \right]. \tag{4}$$

For the **RL objective**, models operate without labels, directly minimizing the cost of the decoded solution. The generator $\tilde{\pi}_\theta$ is trained to minimize the expectation of the cost $c_g(f_g(\tilde{\boldsymbol{x}}))$ subsequent to decoding via $f_g$:

$$\mathcal{R}(\theta) = \mathbb{E}_{g \sim \mathcal{G}, \tilde{\boldsymbol{x}} \sim \tilde{\pi}_\theta(\cdot|g)} \left[ -c_g(f_g(\tilde{\boldsymbol{x}})) \right]. \tag{5}$$

Notably, the RL objective $\mathcal{R}(\theta)$ directly optimizes the NCO goal of cost minimization outlined in (2). However, it often yields lower performance compared to SL-based counterparts across CO benchmarks (Qiu et al., 2022; Sun & Yang, 2023; Li et al., 2023; 2024). Conversely, while the SL objective $\mathcal{L}(\theta)$ serves as a surrogate, it has been widely adopted due to its stronger empirical performance relative to RL alternatives; in principle, if $\tilde{\pi}_\theta$ perfectly imitates the optimal labels, it inherently fulfills the fundamental CO goal as defined in (1). Driven by this empirical advantage, the SL paradigm has generally been preferred over RL for heatmap-based solvers.

## 2.3 Diffusion Models for Combinatorial Optimization

Discrete diffusion models have emerged as a powerful class of generative models (Austin et al., 2021), recently achieving state-of-the-art results among heatmap-based solvers for CO problems (Sun & Yang, 2023). Since these models are trained to mimic the distribution of a given solution dataset by minimizing the SL objective

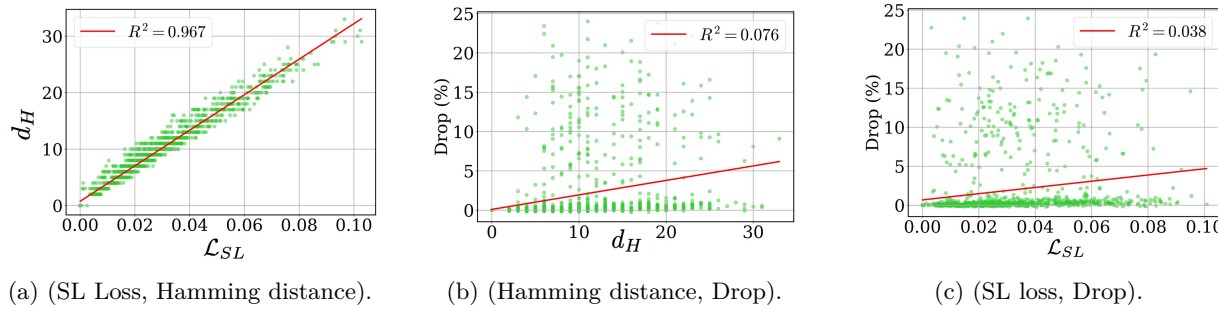

(a) (SL Loss, Hamming distance).     (b) (Hamming distance, Drop).     (c) (SL loss, Drop).

Figure 1: Scatter plots and correlation analysis for H1 and H2 on TSP-100. (a) Surrogate loss $\mathcal{L}_{SL}$ vs. Hamming distance $d_{\mathrm{H}}$ (edge disagreements). (b) Hamming distance $d_{\mathrm{H}}$ vs. Drop (% cost gap to $\boldsymbol{x}^\star$). (c) Surrogate loss $\mathcal{L}_{SL}$ vs. Drop (% cost gap to $\boldsymbol{x}^\star$).

$\mathcal{L}(\theta)$ defined in (4), they are fundamentally classified as SL-based heatmap solvers. The framework operates via a fixed forward process and a learned reverse process. The forward process systematically corrupts a ground-truth solution, $\mathbf{x}_0 = \boldsymbol{x}_g^\star$, by injecting noise over $T$ timesteps to produce a purely random vector $\mathbf{x}_T$. The reverse process is trained to iteratively denoise $\mathbf{x}_T$ back to the original solution $\mathbf{x}_0$, effectively functioning as the heatmap generator $\tilde{\pi}_\theta(\cdot|g)$. A detailed formulation is provided in the Appendix A.

## 3   Empirical Analysis of Objective Mismatch

Comparing the formulations of the SL objective $\mathcal{L}(\theta)$ and the RL objective $\mathcal{R}(\theta)$ reveals a fundamental disconnect. While $\mathcal{R}(\theta)$ explicitly optimizes the post-decoded cost $c_g(f_g(\tilde{\boldsymbol{x}}))$, $\mathcal{L}(\theta)$ *remains blind to* the non-differentiable decoder $f_g$ and the true cost function $c_g$—what we term **Decoder-Blindness** and **Cost-Blindness**, respectively. The SL paradigm has been widely adopted under the implicit hypothesis that heatmaps closely approximating optimal solutions would naturally yield low-cost decoded solutions after decoding. We empirically test whether this hypothesis holds in practice, using a well-trained state-of-the-art SL-based heatmap solver, DIFUSCO (Sun & Yang, 2023). Specifically, the SL paradigm tacitly relies on the following two hypotheses:

(H1) **Decoder Monotonicity:** A reduction in the SL loss $\mathcal{L}_{SL}$ in (4) leads to a decoded solution $f_g(\tilde{\boldsymbol{x}})$ that is structurally closer to the optimal solution $\boldsymbol{x}^\star$. Here, $d_H(\boldsymbol{x}, \boldsymbol{x}^\star)$ measures the **Hamming distance**, i.e., the number of mispredicted elements (edges for TSP, nodes for MIS), which measures element-wise disagreement between two discrete solutions, consistent with the element-wise nature of $\mathcal{L}_{SL}$. Formally:

$$\mathcal{L}_{SL}(\tilde{\boldsymbol{x}}_A, \boldsymbol{x}^\star) < \mathcal{L}_{SL}(\tilde{\boldsymbol{x}}_B, \boldsymbol{x}^\star) \implies d_H(f_g(\tilde{\boldsymbol{x}}_A), \boldsymbol{x}^\star) < d_H(f_g(\tilde{\boldsymbol{x}}_B), \boldsymbol{x}^\star). \tag{6}$$

(H2) **Cost Smoothness:** A higher structural similarity to the optimal solution $\boldsymbol{x}^\star$ directly translates into a lower solution cost, the ultimate objective of CO. Formally, for any two decoded solutions $\boldsymbol{x}_A$ and $\boldsymbol{x}_B$:

$$d_H(\boldsymbol{x}_A, \boldsymbol{x}^\star) < d_H(\boldsymbol{x}_B, \boldsymbol{x}^\star) \implies c_g(\boldsymbol{x}_A) < c_g(\boldsymbol{x}_B). \tag{7}$$

If both hypotheses held universally, $\mathcal{L}(\theta)$ would be a valid objective for $\mathcal{R}(\theta)$. In Figure 1, we empirically test these hypotheses. Regarding H1, Figure 1a shows a strong but imperfect correlation—accurate heatmaps do not always yield solutions structurally close to the optimum. More critically, contradicting H2, Figure 1b reveals a near-zero correlation between this structural proximity and the final solution cost. This disconnect reflects the complex landscape of CO, where structural proximity to the optimal solution bears little relation to the solution cost. Collectively, these findings reveal a notable limitation of the imitation-centric SL paradigm for heatmap-based solvers, underscoring the benefit of a training framework that explicitly incorporates both decoder behavior $f_g$ and true cost feedback $c_g$. Figure 1c directly examines the correlation between the surrogate loss $\mathcal{L}_{SL}$ and the actual solution cost (Drop), revealing the weakest relationship among all three plots ($R^2 = 0.038$). This result is consistent with the individual findings on Decoder-Blindness (H1)

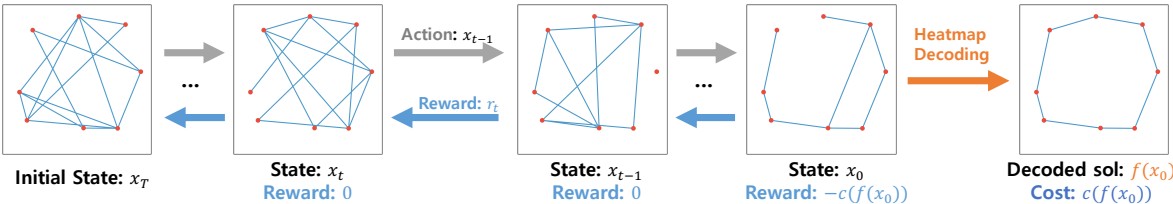

Figure 2: The denoising process formulated as an MDP with initial noise $\mathbf{x}_T \sim \text{Bern}(\boldsymbol{p} = 0.5^N)$.

and Cost-Blindness (H2), suggesting that, for a well-trained SL-based heatmap solver, further reducing the surrogate loss provides little benefit in lowering the actual solution cost.

# 4 Methods

We propose an RL fine-tuning framework that aligns a pre-trained diffusion model $\pi_{\theta_{SL}}$, trained with the SL objective $\mathcal{L}(\theta)$, with the true CO objective. Our goal is to fine-tune the pre-trained diffusion model $\pi_{\theta_{SL}}$ via RL with the cost-aware objective $\mathcal{R}(\theta)$, yielding $\pi_{\theta_{RL}}$. In this work, we use the publicly available DIFUSCO (Sun & Yang, 2023) checkpoints as our SL initialization.

## 4.1 MDP Formulation for Cost-Aware Training

To address the limitations of the SL objective from Decoder- and Cost-Blindness, we formulate the denoising process as a Markov Decision Process (MDP) that integrates cost $c_g$ and decoder $f_g$ into the fine-tuning process of heatmap-based diffusion solvers. The MDP is represented by the tuple $(\mathcal{S}, \mathcal{A}, P, \rho_0, R)$. Here, $\mathbf{s} \in \mathcal{S}$ represents a state in the state space $\mathcal{S}$, and $\mathbf{a} \in \mathcal{A}$ denotes an action in the action space $\mathcal{A}$. The state transition distribution is given by $P(\mathbf{s}_{t+1} \mid \mathbf{s}_t, \mathbf{a}_t)$, $\rho_0(\mathbf{s}_0)$ defines the initial state distribution, and $R(\mathbf{s}_t, \mathbf{a}_t)$ represents the reward function. The objective of reinforcement learning is to train the heatmap generator $\tilde{\pi}_\theta$ that maximizes the cumulative sum of rewards.

$$
\begin{aligned}
\mathbf{s}_t &\triangleq (g, T - t, \mathbf{x}_{T-t}), & \mathbf{a}_t &\triangleq \mathbf{x}_{t-1}, \\
\tilde{\pi}_\theta(\mathbf{a}_t \mid \mathbf{s}_t) &\triangleq p_\theta(\mathbf{x}_{t-1} \mid \mathbf{x}_t, g), & P(\mathbf{s}_{t+1} \mid \mathbf{s}_t, \mathbf{a}_t) &\triangleq (\delta_g, \delta_{T-t-1}, \delta_{\mathbf{x}_{T-t-1}}), \\
\rho_0(\mathbf{s}_T) &\triangleq \left(g, T, \text{Bern}(\boldsymbol{p} = 0.5^{N_g})\right), & R(\mathbf{s}_t, \mathbf{a}_t) &\triangleq \begin{cases} -c_g(f_g(\mathbf{x}_0)) & \text{if } t = T, \\ 0 & \text{otherwise.} \end{cases}
\end{aligned}
\tag{8}
$$

We define $\text{Bern}(\mathbf{p})$ as a Bernoulli distribution with vector probabilities $\mathbf{p}$ for sampling the initial noise $\mathbf{x}_T$, and $\delta_y$ denotes the Dirac delta distribution centered at $y$. To train the heatmap generator $\tilde{\pi}_\theta$, we apply REINFORCE (Williams, 1992) to optimize the iterative denoising procedure using the learning objective:

$$
\nabla_\theta \mathcal{R}(\theta) = \mathbb{E}\left[\sum_{t=1}^{T} \nabla_\theta \log \tilde{\pi}_\theta\left(\mathbf{x}_{t-1} \mid \mathbf{x}_t, g\right) R_t\right] \text{ where } R_t = -c_g\left(f_g(\mathbf{x}_0)\right).
\tag{9}
$$

By explicitly incorporating both the non-differentiable decoder $f_g$ and the true cost $c_g$ into the reward, our framework directly addresses both **Decoder-Blindness** and **Cost-Blindness**, while benefiting from the training stability provided by the SL initialization. Our framework assumes that the cost function $c_g(\cdot)$ can be evaluated exactly in polynomial time. This assumption holds for the standard CO problems studied in the NCO literature (e.g., TSP, MIS, CVRP, Knapsack), since NP problems by definition have solutions verifiable in polynomial time. We note that when $T = 1$, this formulation reduces to a single-step MDP, making our framework model-agnostic and applicable to general SL-based heatmap solvers beyond diffusion models (see Appendix D).

## 4.2 Standard and Label-Centered Reward Strategies

We investigate two distinct reward formulation strategies for CADO: the **Standard Reward (SR)**, which adopts the canonical reward formulation in NCO, and the **Label-Centered Reward (LCR)**, a novel design that strategically exploits the cost information from the pre-training data.

**Standard Reward (SR).**   Our default approach utilizes the negative of the true solution cost directly as the reward signal. To stabilize training and reduce variance, we apply standard batch-wise reward normalization. Crucially, this label-free formulation enables self-sufficient learning through trial-and-error exploration, offering a practical alternative to LCR when labeled solutions are limited while maintaining competitive performance.

**Label-Centered Reward (LCR).**   We propose the Label-Centered Reward (LCR) to further leverage the valuable information embedded in the SL dataset. Rather than direct imitation—which suffers from the objective mismatch discussed earlier—LCR repurposes the *ground-truth solution cost $b_{\mathcal{D}}(g)$ as an instance-specific baseline*. We define the label-centered reward function as the negative optimality gap:

$$R_t = -\left(c_g(f_g(\mathbf{x}_0)) - b_{\mathcal{D}}(g)\right). \tag{10}$$

Notably, since the baseline $b_{\mathcal{D}}(g)$ depends solely on the problem instance $g$ and is independent of the policy's actions, it remains a theoretically *unbiased baseline*, ensuring that the policy gradient remains consistent with the true cost minimization objective, regardless of label optimality. This property is practically significant in CO, where acquiring large-scale, optimal training datasets is often computationally prohibitive due to NP-hardness. In this work, we evaluate both variants; for LCR, we strictly reuse the existing SL pre-training dataset to ensure a fair comparison.

## 4.3 LoRA with Selective Layer Fine-Tuning

To achieve parameter-efficient and stable RL fine-tuning, we employ a hybrid training strategy, **Hybrid Fine-Tuning (Hybrid-FT)**, for our diffusion model architecture comprising an input layer, 12 Graph Neural Network (GNN) (Bresson & Laurent, 2018) layers, and an output layer. We apply **Low-Rank Adaptation (LoRA)** (Hu et al., 2022) to the input layer and the first 11 GNN layers, preserving robust pre-trained features by introducing only a small set of trainable low-rank matrices. For layers most critical to final heatmap generation—the final GNN layer and the output layer—we employ **Selective Layer Fine-Tuning (Selective-FT)**, retraining all parameters in these layers. This hybrid approach enhances training stability and memory efficiency without performance degradation.

## 5 Related Work

**Neural Combinatorial Optimization Solvers.**   Neural approaches to CO generally fall into two categories: autoregressive and heatmap-based solvers. Autoregressive solvers construct solutions sequentially (Kool et al., 2019; Kwon et al., 2020; Kim et al., 2022; Chalumeau et al., 2023; Meng et al., 2025; Liao et al., 2025; Fang et al., 2025; Chalumeau et al., 2025), a process amenable to RL but constrained by high inference latency as problem size scales. In contrast, heatmap-based solvers generate a solution in a one-shot manner, enabling highly efficient parallel inference. While early attempts to train these solvers using RL (Qiu et al., 2022) or unsupervised learning (Min et al., 2023; Sanokowski et al., 2024) often yielded limited performance gains for large-scale CO, recent supervised learning (SL) approaches utilizing powerful generative models—such as diffusion models (Graikos et al., 2022; Sun & Yang, 2023; Wang et al., 2025)—have achieved state-of-the-art results. Other lines of work explore divide-and-conquer (Ye et al., 2024; Zheng et al., 2024) and destroy-and-repair (Li et al., 2025) strategies for large-scale problems.

**The Objective Mismatch in SL-Based Heatmap Solvers.**   Despite their success, SL-based heatmap solvers face an intrinsic limitation: they are trained to imitate optimal solutions rather than to minimize solution cost. Recent works have begun to scrutinize this limitation. SoftDIST (Xia et al., 2024) highlighted the mathematical mismatch between SL and RL objectives, providing indirect evidence of suboptimal performance by showing that complex diffusion models often perform comparably to simple rule-based heatmap generators. To mitigate this, T2T (Li et al., 2023) and FastT2T (Li et al., 2024) introduced inference-time cost guidance,

steering the denoising process toward lower-cost regions. However, these post-hoc methods do not fully achieve true objective alignment: the non-differentiability of the decoder requires them to guide diffusion with a differentiable surrogate cost function, and their effectiveness remains heavily dependent on the quality of the underlying pre-trained model. Our work differs from prior studies in two critical aspects. First, whereas previous research primarily observed the mathematical discrepancy between objectives, we empirically characterize this mismatch and show that it leads to performance degradation. Second, we propose a principled resolution to this objective mismatch via CADO, a reinforcement learning fine-tuning framework.

**RL Fine-Tuning for Diffusion Models.** RL fine-tuning for diffusion models has largely focused on text-to-image generation (Fan et al., 2023; Lee et al., 2023; Wallace et al., 2024), utilizing learned rewards and KL regularization to align with human preferences. We diverge from this paradigm by applying RL fine-tuning to diffusion-based CO solvers. A key advantage of CO over image synthesis is the availability of exact cost functions and deterministic decoders. This allows us to optimize the true CO objective directly, eliminating reliance on potentially unstable learned rewards or KL constraints. Moreover, we propose *Label-Centered Reward* for better utilization of ground-truth data and *Hybrid Fine-Tuning* to enable the efficient adaptation of large-scale GNNs.

## 6 Experiment

The experiments are conducted using eight NVIDIA L40 GPUs for training and one L40 GPU for testing, along with an AMD EPYC 7413 24-Core Processor.

### 6.1 Experiment Settings

**Problems and Datasets.** We evaluate CADO on widely adopted benchmarks for the Traveling Salesman Problem (TSP) and Maximum Independent Set (MIS). For TSP, we utilize standard datasets ranging from 100 to 10k nodes. For MIS, we employ the SATLIB (MIS-SAT) and Erdős-Rényi (MIS-ER) graph benchmarks. All experiments follow standard train/test protocols established in prior works (Kool et al., 2019; Fu et al., 2021; Sun & Yang, 2023).

**Evaluation Metrics.** We assess our model and other baselines using three metrics. (1) **Cost**: For TSP, we measure the average tour length (lower is better). For MIS, we measure the average size of the independent set (higher is better). (2) **Drop**: We calculate the average drop (optimality gap) between the model-generated solutions and optimal solutions. (3) **Time**: We record the total runtime during testing.

**Inference Strategies.** To transform heatmaps into feasible solutions, we follow the conventional decoding and post-hoc refinement protocols adopted by prevailing heatmap-based solvers (Qiu et al., 2022; Sun & Yang, 2023; Li et al., 2023; Wang et al., 2025). For MIS, we utilize a greedy decoder without further refinement. For TSP, we utilize a greedy decoder and optionally apply post-hoc refinement using the 2-opt heuristic (Lin & Kernighan, 1973) or Monte Carlo Tree Search (MCTS). Furthermore, we incorporate the Local Rewrite (LR) strategy to iteratively enhance solution quality via partial noise injection and reconstruction. Crucially, unlike T2T (Li et al., 2023) which relies on gradient-based cost-guided search (CS) during rewriting, our method operates without such auxiliary guidance, relying solely on the robust heatmap learned via RL fine-tuning. We also evaluate **CADO-L**, an ablated version of CADO that excludes LR. This variant enables a direct evaluation of our RL fine-tuning, isolating the effect of additional search techniques, while *reducing computational overhead to 40%* of baseline methods. All CADO results are the average of 4 independent runs.

**Baselines.** We compare our method with the following methods: (1) **Exact Solvers**: Concorde (Applegate et al., 2006) and Gurobi (Gurobi Optimization, LLC, 2020); (2) **Heuristics**: LKH3 (Helsgaun, 2017), KaMIS (Lamm et al., 2016), Farthest Insertion; (3) **Supervised learning (SL):** GCN (Joshi et al., 2019), BQ (Drakulic et al., 2023), LEHD (Luo et al., 2023), DIFUSCO (Sun & Yang, 2023), T2T (Li et al., 2023), DEITSP (Wang et al., 2025), FastT2T (Li et al., 2024), DRHG (Li et al., 2025); (4) **Reinforcement learning (RL)**: AM (Kool et al., 2019), POMO (Kwon et al., 2020), DIMES (Qiu et al., 2022), COMPASS (Chalumeau

et al., 2023), ICAM (Zhou et al., 2024), GLOP (Ye et al., 2024), UDC (Zheng et al., 2024), HierTSP (Goh et al., 2024), LRBS (Verdù et al., 2025), MEMENTO (Chalumeau et al., 2025); (5) **Unsupervised learning (UL)** : GFlowNets (Zhang et al., 2023), UTSP (Min et al., 2023).

## 6.2 Addressing the Objective Mismatch in SL via RL

As discussed in Section 3 (see Figure 1), the two blindnesses in the SL objective cause objective mismatch, which can degrade performance. In this section, we empirically evaluate whether our proposed framework can effectively address these blindnesses.

**Cost-Blindness.** Figure 3 tracks both the SL loss and solution cost (Drop) throughout RL fine-tuning. While CADO-L reduces the drop by 58.5% (from 0.6% to 0.25%), the SL loss monotonically increases. Since the SL and RL objectives are not perfectly aligned, some degree of divergence is inherent. However, the sustained nature of this pattern suggests that the two objectives likely have distinct optima, rather than converging to the same optimum through different loss landscapes. This is consistent with the near-zero correlation ($R^2 = 0.038$) between SL loss and solution cost observed in Figure 1c, and the negligible gains from extended SL training (DIFUSCO+SFT in Table 4). Together, these results indicate that explicit objective alignment with cost information is necessary to push performance beyond the SL plateau.

**Decoder-Blindness.** We next validate that CADO's decoder-aware objective indeed overcomes **Decoder-Blindness** in SL (Figure 1a). To demonstrate this, we test the standard SL-trained DIFUSCO with two decoders: a Greedy (`Grdy`) decoder and a simpler Nearest Neighbor (`NN`) decoder. As shown in Table 1, DIFUSCO's performance degrades significantly when switching to the simpler `NN` decoder (1.62% to 2.32%) even though the underlying heatmap remains identical. This implies that the heatmap generation process should be explicitly conditioned on the specific decoder used. In contrast, we train two CADO-L variants, each fine-tuned for one of the decoders. In contrast to DIFUSCO's static heatmap, CADO-L achieves peak performance when the inference decoder matches the one used during training, confirming its ability to learn decoder-specific structural priors. This indicates that CADO-L's integration of decoder feedback generates heatmaps structured for its paired decoder, directly addressing Decoder-Blindness of SL.

## 6.3 Main Results for Varied CO Benchmarks

We evaluate CADO on widely-used benchmarks for the TSP and the MIS. Across all settings, CADO demonstrates consistently strong performance. Notably, this robustness is achieved without extensive, task-specific hyperparameter tuning; we employ a nearly identical configuration for all experiments, adjusting only the number of training epochs. Detailed experimental settings are provided in the Appendix C.

**TSP-100/500/1k/10k.** Table 2 summarizes the results on TSP instances ranging from 100 to 10k nodes. For a fair comparison, we standardized the computational budget for inference: autoregressive baselines were adjusted to match CADO's inference time, and 2-opt for heatmap-based solvers was capped at 1,000 iterations. Across all scales, CADO consistently outperforms the majority of baselines. Notably on TSP-1k,

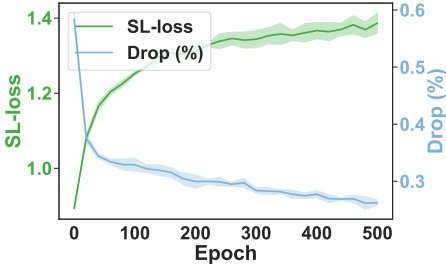

Figure 3: Learning curve of CADO-L for MIS-SAT. The average of 4 independent runs.

| Algorithm | Test (`Grdy`) Drop ↓ | Test (`NN`) Drop ↓ |
|---|---|---|
| DIFUSCO | 1.62% | 2.32% |
| CADO-L (`Grdy`) | **0.19%** | 0.34% |
| CADO-L (`NN`) | 0.31% | **0.31%** |

Table 1: Impact of decoder-aware training.

| Algorithm | Type | TSP-100 | | | TSP-500 | | | TSP-1k | | | TSP-10k | | |
|---|---|---|---|---|---|---|---|---|---|---|---|---|---|
| | | Length ↓ | Drop ↓ | Time | Length ↓ | Drop ↓ | Time | Length ↓ | Drop ↓ | Time | Length ↓ | Drop ↓ | Time |
| Concorde | Exact | 7.76* | - | 40m | 16.55* | - | 38m | 23.12* | - | 6.7h | - | - | - |
| LKH-3 | Heuristics | 7.76 | 0.00% | 1.4h | 16.55 | 0.00% | 46m | 23.12 | 0.00% | 2.6h | 71.77* | - | 8.8h |
| FI | Heuristics | 8.72 | 12.36% | 0s | 18.30 | 10.57% | 0s | 25.72 | 11.25% | 0s | 80.59 | 12.29% | 6s |
| AM | RL+BS | 7.95 | 4.53% | 6s | 20.02 | 20.99% | 1.5m | 31.15 | 34.8% | 3.2m | 141.68 | 97.4% | 6.0m |
| EAN | RL+2-opt | - | - | - | 23.75 | 43.6% | 58m | 47.73 | 106% | 5.4h | - | - | - |
| POMO | RL | 7.84 | 1.07% | 2s | 19.24 | 16.25% | 13h | - | - | - | - | - | - |
| BQ† | SL | 7.79 | 0.35% | - | 16.72 | 1.18% | 0.8m | 23.65 | 2.27% | 1.9m | - | - | - |
| LEHD | SL+RRC20 | 7.76 | 0.04% | 0.4m | 16.66 | 0.66% | 1.6m | 23.51 | 1.70% | 11m | - | - | - |
| ICAM† | RL+Aug8 | 7.77 | 0.15% | 37s | 16.55 | 0.77% | 38s | 23.49 | 1.58% | 3.8m | - | - | - |
| SIL | RL+PRC50 | - | - | - | - | - | - | 23.46 | 1.50% | 9.0m | 73.77 | 2.78% | 10m |
| DRHG | SL+DR150 | **7.76** | **0.00%** | 9.0m | 16.65 | 0.62% | 2.9m | 23.55 | 1.85% | 3.1m | - | - | - |
| LRBS† | RL | 7.76 | 0.01% | 22h | 17.19 | 3.97% | 2.0h | 25.90 | 11.89% | 8.1h | - | - | - |
| COMPASS† | RL | - | - | - | 16.81 | 1.60% | - | - | - | - | - | - | - |
| MEMENTO† | RL | - | - | - | 16.80 | 1.59% | - | - | - | - | - | - | - |
| GLOP | RL+DC | - | - | - | 16.91 | 1.99% | 1.5m | 23.84 | 3.11% | 3.0m | 75.29 | 4.90% | 1.8m |
| UDC† | RL+DC | - | - | - | 16.78 | 1.58% | 4.0m | 23.53 | 1.78% | 8.0m | - | - | - |
| GCN | SL | 8.41 | 8.38% | 6m | 29.72 | 79.6% | 6.7m | 48.62 | 110% | 29m | - | - | - |
| UTSP† | UL+MCTS | **7.76** | **0.00%** | 1.1m | 17.11 | 3.41% | 3.0m | 24.14 | 4.40% | 6.7m | - | - | - |
| SoftDIST | MCTS | - | - | - | 16.78 | 1.44% | 1.7m | 23.63 | 2.20% | 3.3m | 74.03 | 3.13% | 17m |
| DIMES | RL+MCTS | - | - | - | 16.87 | 1.93% | 2.9m | 23.73 | 2.64% | 6.9m | 74.63 | 3.98% | 30m |
| DEITSP†‡ | SL+2-opt | 7.77 | 0.10% | 4.0m | 16.90 | 2.15% | 9.3m | 23.97 | 3.68% | 42m | - | - | - |
| FastT2T | SL+CS+2-opt | 7.76 | 0.01% | 1.6m | 16.66 | 0.65% | 46s | 23.35 | 0.99% | 3.5m | - | - | - |
| DIFUSCO | SL+2-opt | 7.78 | 0.41% | - | 16.81 | 1.55% | 5.8m | 23.55 | 1.86% | 18m | 73.99 | 3.10% | 35m |
| T2T | SL+CS+2-opt | 7.76 | 0.06% | - | 16.68 | 0.83% | 4.8m | 23.41 | 1.26% | 18m | - | - | - |
| CADO | SL+RL+2-opt | 7.76 | 0.06% | 5.4m | **16.65** | **0.61%** | 1.7m | **23.32** | **0.88%** | 3.6m | **73.69** | **2.68%** | 13m |

Table 2: Results on TSP-100/500/1k/10k. BS: Beam Search, RRC: Random Re-Construct, CS: Cost-guided Search, DC: Divide-and-Conquer. * represents the baseline for computing the drop. The results of models marked with † are evaluated on different test datasets and are taken from their respective papers. ‡ denotes generalization results from models trained on a different distribution.

while the SL-trained DIFUSCO (1.86%) shows marginal improvement over the non-learned SoftDIST (2.20%), CADO achieves a significantly reduced drop of 0.88%, effectively halving the drop of DIFUSCO. On TSP-10k, despite being a one-shot heatmap generator, CADO surpasses divide-and-conquer methods (e.g., GLOP, UDC) explicitly designed for large-scale efficiency.

**MIS-SAT/ER.** Our approach demonstrates strong performance on MIS problems, outperforming state-of-the-art baselines across two distinct graph distributions: MIS-SAT and MIS-ER. These widely adopted benchmarks inherently present disparate pre-training qualities: the SL baseline achieves high accuracy on MIS-SAT (0.33% drop) but fails significantly on MIS-ER (18.53% drop).

Despite this large disparity in initial model quality, CADO demonstrates consistent robustness. While it refines the already strong MIS-SAT model, its impact is most significant in the challenging MIS-ER regime. Here, CADO achieves a drop of only 2.78%, representing an 85% gap reduction from the pretrained DIFUSCO (18.53%) and a 70% improvement over the previous best method, FastT2T (9.52%). This substantial margin underscores the effectiveness of correcting the objective mismatch, and empirically corroborates our analysis in Section 3: the objective mismatch manifests most severely when heatmap-based SL solvers yield highly suboptimal heatmaps, an issue that CADO mitigates through its objective alignment.

### 6.4 Detailed Analysis of Heatmap-Based Solvers

This section dissects the core methodologies of heatmap-based solvers, distinguishing between SL, RL, post-hoc cost guidance, and our proposed RL fine-tuning. To rigorously isolate the intrinsic efficacy of each learning objective, we introduce two control baselines: (1) **RL-Scratch**, trained from scratch to evaluate the necessity of SL initialization, and (2) **DIFUSCO+SFT**, which extends SL training by 50% additional epochs to ensure that CADO's improvements are not merely due to extended training. We disable all auxiliary search heuristics (e.g., 2-opt, MCTS) to ensure a controlled comparison of the underlying heatmap-based solvers.

Pure RL approaches, DIMES and RL-Scratch, exhibit significantly inferior performance compared to SL-based counterparts. This result underscores that SL pre-training serves as a requisite warm-start for navigating

| Algorithm | Type | SATLIB | | | ER-[700-800] | | |
|---|---|---|---|---|---|---|---|
| | | Size ↑ | Drop ↓ | Time | Size ↑ | Drop ↓ | Time |
| KaMIS | Heuristics | 425.96* | - | 38m | 44.87* | - | 52m |
| Gurobi | Exact | 425.95 | 0.00% | 26m | 41.28 | - | 50m |
| LwD | RL+S | 422.22 | 0.88% | 19m | 41.17 | 8.25% | 6.3m |
| GFlowNets | UL+S | 423.54 | 0.57% | 23m | 41.14 | 8.53% | 2.9m |
| UDC | RL+DC | - | - | - | 42.88 | 4.44% | 21m |
| Intel | SL | 420.66 | 1.48% | 23m | 34.86 | 22.31% | 6.1m |
| DIMES | RL | 421.24 | 1.11% | 24m | 38.24 | 14.78% | 6.1m |
| DIFUSCO | SL | 424.56 | 0.33% | 8.3m | 36.55 | 18.53% | 8.8m |
| T2T | SL+CS | 425.02 | 0.22% | 8.1m | 39.56 | 11.83% | 8.5m |
| FastT2T | SL+CS | - | - | - | 40.60 | 9.52% | 41s |
| CADO | SL+RL | **425.43** | **0.12%** | 6.5m | **43.62** | **2.78%** | 1.8m |

Table 3: Results on SATLIB and ER-[700-800].

the high-dimensional combinatorial landscape. On the other hand, the SL-based DIFUSCO+SFT yields negligible improvements over DIFUSCO and remains significantly inferior to cost-integrated baselines. This performance plateau implies that the sub-optimality of SL solvers is not merely a symptom of insufficient training steps, but a consequence of the fundamental objective mismatch inherent in the SL objective itself.

In comparing cost integration strategies, CADO-L outperforms T2T and FastT2T even in its standalone form without Local Rewrite (LR), while these baselines require LR despite its additional computational overhead. When LR is applied to CADO as well, the performance gap widens further. These results indicate that fundamentally realigning the generative prior via RL fine-tuning is more effective than applying post-hoc guidance via surrogate cost functions. We provide a more comprehensive comparison with other cost-aware heatmap-based solvers in Appendix D.

We also validate CADO's framework on other SL-based heatmap solvers in Appendix D. Crucially, CADO operates as a model-agnostic framework capable of enhancing various heatmap-based SL solvers beyond diffusion models. For instance, applying CADO to the pre-trained FastT2T (1,0), a general heatmap solver that operates via a single forward pass rather than a diffusion process, yields substantial performance gains, as detailed in Tables 13 and 14. This shows that our framework is a versatile tool that effectively resolves the objective mismatch for a wide range of heatmap-based solvers.

## 6.5 Ablation Study

**Impact of Reward Strategies.** We compare CADO with SR and LCR in Table 5. The results reveal two key insights. First, both variants substantially outperform the original DIFUSCO and extended supervised fine-tuning (DIFUSCO+SFT). Notably, even the label-free CADO (w/ SR) alone strongly outperforms all baselines, confirming that RL-based objective alignment is the primary driver of performance. Second, CADO

| Algorithm | Type | TSP-500 Drop ↓ | TSP-1k Drop ↓ | MIS-SAT Drop ↓ | MIS-ER Drop ↓ |
|---|---|---|---|---|---|
| DIMES | RL | 15.0% | 15.0% | 1.1% | 14.8% |
| RL-Scratch | RL | 18.0% | 17.5% | 0.88% | 26.4% |
| DIFUSCO | SL | 11.2% | 11.2% | 0.33% | 18.5% |
| DIFUSCO+SFT | SL | 10.1% | 12.3% | 0.29% | 12.4% |
| T2T | SL+CS+LR | 6.9% | 9.8% | 0.22% | 11.8% |
| FastT2T (5,3) | SL+CS+LR | 5.5% | 8.9% | - | 9.5% |
| CADO-L | SL+RL | **3.0%** | **6.1%** | **0.16%** | **4.3%** |
| CADO | SL+RL+LR | **1.3%** | **4.0%** | **0.12%** | **2.8%** |

Table 4: Comparisons between different cost information integration strategies for Heatmap-Based Solvers. SL/RL stands for Supervised Learning and Reinforcement Learning, respectively. CS stands for cost-guided search. LR stands for Local Rewrite.

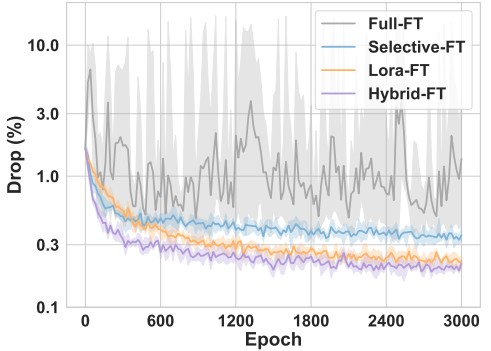

Figure 4: Learning curve of various fine-tuning methods. The result is the average of 4 independent runs.

| Algorithm | TSP-100 Drop ↓ | TSP-500 Drop ↓ | MIS-SAT Drop ↓ |
|---|---|---|---|
| DIFUSCO | 1.01% | 11.2% | 0.33% |
| DIFUSCO+SFT | 1.05% | 10.1% | 0.29% |
| CADO-L (w/ SR) | **0.22%** | **3.3%** | **0.19%** |
| CADO-L (w/ LCR) | **0.19%** | **3.0%** | **0.16%** |
| Improvement of LCR (%) | 15.8% | 10.0% | 18.8% |

Table 5: Ablation Study for Label-Centered Reward.

(w/ LCR) yields an additional 10.0%–18.8% improvement over CADO (w/ SR). This shows that leveraging ground-truth labels as an unbiased baseline better aligns the CO objective rather than mere imitation.

**Analysis of LoRA and Selective Layer Fine-Tuning.** We compare several parameter-efficient fine-tuning strategies on the TSP-100. As illustrated in Figure 4, naively fine-tuning all parameters (Full-FT) proves highly unstable. Applying only LoRA (LoRA-FT) ensures stability but suffers from slow initial convergence. Conversely, fine-tuning only the final, selective layers (Selective-FT) yields rapid initial gains but quickly plateaus due to limited expressive power to adapt the entire model. Our Hybrid-FT strategy, which judiciously combines LoRA for backbone layers and full fine-tuning for final layers, resolves these trade-offs. It achieves consistently superior results, demonstrating both fast initial convergence and the best final performance. These results validate Hybrid-FT as a practical RL fine-tuning approach that maintains both stability and strong expressive power for large-scale GNNs. The learning curves of CADO with Hybrid-FT across all five CO tasks are provided in Appendix F.

## 6.6 Analysis on Robustness and Generalization

### 6.6.1 Generalization to Real-World Dataset

To evaluate the generalization ability of CADO to unseen real-world scenarios, we test the TSP-100-trained model on **TSPLIB (50–200 nodes)**, a widely used real-world TSP benchmark. As shown in Table 6, CADO outperforms other baselines with a 0.46% drop, reducing the gap by approximately 41% compared to the previous best. To further validate CADO in a larger-scale setting, we also test the TSP-500-trained model on **TSPLIB (200–1000 nodes)**, where CADO again achieves the highest score among diffusion-based solvers. Detailed results for all TSPLIB benchmarks are provided in Table 16 in Appendix E.

### 6.6.2 Performance Under Low-Quality Training Data

We evaluate the robustness of heatmap-based solvers using a suboptimal TSP-100 dataset generated by a time-limited LKH solver, which exhibits a known 1.36% drop. This setting mirrors practical scenarios where obtaining optimal solutions for large-scale CO problems is computationally intractable, necessitating the use of suboptimal solutions. As shown in Table 7, imitation-centric methods suffer severely from data quality degradation. DIFUSCO exhibits error amplification, deteriorating to an 11.84% gap, while even T2T's post-hoc guidance fails to fully mitigate the flawed pre-training (6.99%).

In contrast, CADO demonstrates strong robustness. CADO (w/ SR) achieves a drop of only 1.86%, significantly outperforming all other baselines. Notably, leveraging suboptimal labels via the Label-Centered Reward further boosts performance to 1.71%. Notably, the Label-Centered Reward remains superior to the Standard Reward even when the labels are noisy. This empirically aligns with our theoretical claim in Section 4.2 that

| Algorithm | TSPLIB (50–200) | |
| --- | --- | --- |
| | Drop ↓ | Time |
| Concorde | 0.00% | - |
| AM | 3.93% | 13.2s |
| POMO | 1.39% | 2.9s |
| Sym-NCO | 1.92% | 3.0s |
| ELG | 1.14% | 6.7s |
| HierTSP | 1.78% | 3.9s |
| DIFUSCO | 1.28% | 20.6s |
| T2T | 0.90% | 29.0s |
| DEITSP | 0.78% | 8.5s |
| CADO | **0.46%** | 22.1s |

Table 6: Generalization results on TSPLIB (50–200).

| Algorithm | Drop ↓ |
| --- | --- |
| DIFUSCO | 11.84% |
| T2T | 6.99% |
| CADO (w/ SR) | **1.86%** |
| CADO (w/ LCR) | **1.71%** |

Table 7: Robustness under low-quality training on TSP-100 without the 2-opt heuristic.

the Label-Centered Reward can serve as an unbiased baseline even with suboptimal labels, confirming its practical utility in real-world scenarios.

# 7 Conclusion

In this work, we empirically characterize the objective mismatch—manifested as Decoder- and Cost-Blindness—and show that it acts as a performance bottleneck in supervised learning for heatmap-based CO solvers. To address this, we propose CADO (**C**ost-**A**ware **D**iffusion models for **O**ptimization), a reinforcement learning fine-tuning framework that explicitly realigns the diffusion process with the true CO objective. We introduce Label-Centered Reward, a novel reward formulation for CO that repurposes pre-training labels to ensure effective objective alignment rather than mere imitation. Additionally, our Hybrid Fine-Tuning strategy facilitates the stable and efficient adaptation of large-scale GNN architectures. Extensive experiments show that CADO achieves a new state-of-the-art across TSP and MIS benchmarks while exhibiting robustness to suboptimal datasets.

# 8 Limitations

CADO requires an SL-pretrained solver as its starting point, inheriting the need for a labeled dataset and additional computational cost on top of RL fine-tuning. Moreover, although CADO consistently improves SL-based heatmap solvers across all evaluated tasks, its final performance is influenced by the pre-trained model, as shown by the RL-Scratch results (Table 4).

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

## A    Training objective in Diffusion Model

In this section, we describe the supervised learning approach for training a diffusion-based heatmap solver $\tilde{\pi}_\theta(\boldsymbol{x}|g)$, following the methodology established by Sun & Yang (2023). The diffusion process consists of a forward noising procedure and a reverse denoising procedure. During the forward process, noise is gradually added to the initial solution until it is fully transformed into random noise, creating a sequence of latent variables $\mathbf{x_0}, \mathbf{x_1}, \ldots, \mathbf{x_T}$ where $\mathbf{x_0} = \boldsymbol{x}_g^\star$ in CO and $\mathbf{x}_T$ is completely random noise. The forward noising process is defined by $q(\mathbf{x_{1:T}}|\mathbf{x_0}) = \prod_{t=1}^{T} q(\mathbf{x_t}|\mathbf{x_{t-1}})$. Then, during the reverse denoising procedure, a model is trained to denoise this random noise $\mathbf{x_T}$ back to the high-quality solution $\mathbf{x_0}$. The reverse process is modeled as $\tilde{\pi}_\theta(\mathbf{x_{0:T}}|g) = \tilde{p}(\mathbf{x_T}) \prod_{t=1}^{T} \tilde{\pi}_\theta(\mathbf{x_{t-1}}|\mathbf{x_t}, g)$, with $\theta$ representing the model parameters, and this reverse model is later used as a heatmap-based solver.

The training objective is to match $p_\theta(\mathbf{x_0}|g)$ with the high-quality data distribution $q(\mathbf{x_0}|g)$, by minimizing the variational upper bound of the negative log-likelihood:

$$\mathcal{L}(\theta) = \mathbb{E}_q \Big[ -\log \tilde{\pi}_\theta(\mathbf{x_0}|\mathbf{x_1}, g) + \sum_{t=2}^{T} D_{KL}(q(\mathbf{x_{t-1}}|\mathbf{x_t}, \mathbf{x_0}) \| \tilde{\pi}_\theta(\mathbf{x_{t-1}}|\mathbf{x_t}, g)) \Big]. \tag{11}$$

In CO, considering that each entry of the optimization variable $\boldsymbol{x}$ is an indicator of whether to select a node or an edge, each entry can also be represented as a one-hot $\{0, 1\}^2$ when modeled with a Bernoulli distribution. Therefore, for diffusion process, $\boldsymbol{x}$ turns into $N$ one-hot vectors $\mathbf{x_0} \in \{0, 1\}^{N \times 2}$. Then, a discrete

diffusion model (Austin et al., 2021) is utilized. Specifically, at each time step $t$, the process transitions from $\mathbf{x_{t-1}}$ to $\mathbf{x_t}$ defined by:

$$q(\mathbf{x_t}|\mathbf{x_{t-1}}) = \text{Cat}(\mathbf{x_t}; \mathbf{p} = \boldsymbol{x}_{t-1}\mathbf{Q_t}) \tag{12}$$

where the $\text{Cat}(\boldsymbol{x}; \mathbf{p})$ is a categorical distribution over $x \in \{0,1\}^{N\times 2}$ with vector probabilities $\mathbf{p}$ and transition probability matrix $\mathbf{Q_t}$ is:

$$\mathbf{Q_t} = \begin{bmatrix} (1-\beta_t) & \beta_t \\ \beta_t & (1-\beta_t) \end{bmatrix}. \tag{13}$$

Here, $\beta_t$ represents the noise level at time $t$. The t-step marginal distribution can be expressed as:

$$q(\mathbf{x_t}|\mathbf{x_0}) = \text{Cat}(\mathbf{x_t}; \mathbf{p} = \mathbf{x_0}\overline{\mathbf{Q_t}}) \tag{14}$$

where $\overline{\mathbf{Q_t}} = \mathbf{Q_1}\mathbf{Q_2}, \ldots, \mathbf{Q}_t$. To obtain the distribution $q(\mathbf{x_{t-1}}|\mathbf{x_t}, \mathbf{x_0})$ for the reverse process, Bayes' theorem is applied, resulting in:

$$q(\mathbf{x_{t-1}}|\mathbf{x_t}, \mathbf{x_0}) = \text{Cat}\left(\mathbf{x_{t-1}}; \mathbf{p} = \frac{\mathbf{x_t}{\mathbf{Q_t}}^\top \odot \mathbf{x_0}\overline{\mathbf{Q}}_{\mathbf{t-1}}}{\mathbf{x_0}\overline{\mathbf{Q_t}}\mathbf{x_t}^\top}\right). \tag{15}$$

As in (Austin et al., 2021), the neural network responsible for denoising $p_\theta(\tilde{\mathbf{x}}_0|\mathbf{x_t}, g)$ is trained to predict the original data $\mathbf{x_0}$. During the reverse process, this predicted $\tilde{\mathbf{x}}_0$ is used as a substitute for $\mathbf{x_0}$ to compute the posterior distribution:

$$\tilde{\pi}_\theta(\mathbf{x_{t-1}}|\mathbf{x_t}, g) = \sum_{\boldsymbol{x}} q(\mathbf{x_{t-1}}|\mathbf{x_t}, \tilde{\mathbf{x}}_0)\tilde{\pi}_\theta(\tilde{\mathbf{x}}_0|\mathbf{x_t}, g) \tag{16}$$

## B  Neural Network Architecture

Following Sun & Yang (2023), we also utilize an anisotropic graph neural network with edge gating (Bresson & Laurent, 2018) for the backbone network of the diffusion model.

Consider $h_i^\ell$ and $e_{ij}^\ell$ as the features of node $i$ and edge $ij$ at layer $\ell$, respectively. Additionally, let $t$ represent the sinusoidal features (Vaswani et al., 2017) corresponding to the denoising timestep $t$. The propagation of features to the subsequent layer is performed using an anisotropic message-passing mechanism:

$$\hat{e}_{ij}^{\ell+1} = P^\ell e_{ij}^\ell + Q^\ell h_i^\ell + R^\ell h_j^\ell, \tag{17}$$

$$e_{ij}^{\ell+1} = e_{ij}^\ell + \text{MLP}_e(\text{BN}(\hat{e}_{ij}^{\ell+1})) + \text{MLP}_t(t), \tag{18}$$

$$h_i^{\ell+1} = h_i^\ell + \alpha(\text{BN}(U^\ell h_i^\ell + \sum_{j \in N_i} \sigma(\hat{e}_{ij}^{\ell+1}) \odot V^\ell h_j)), \tag{19}$$

where $U^\ell, V^\ell, P^\ell, Q^\ell, R^\ell \in \mathbb{R}^{d\times d}$ are learnable parameters for layer $\ell$, $\alpha$ denotes the ReLU activation function (Krizhevsky, 2010), BN stands for Batch Normalization (Ioffe & Szegedy, 2015), $\sigma$ is the sigmoid activation function, $\odot$ represents the Hadamard product, $N_i$ indicates the neighbors of node $i$, and $\text{MLP}(\cdot)$ refers to a two-layer multi-layer perceptron.

For the Traveling Salesman Problem (TSP), the initial edge features $e_{ij}^0$ are derived from the corresponding values in $x_t$, and the initial node features $h_i^0$ are initialized using the nodes' sinusoidal features. In contrast, for the Maximum Independent Set (MIS) problem, $e_{ij}^0$ are initialized to zero, and $h_i^0$ are assigned values corresponding to $x_t$. We then apply a classification or regression head, with two neurons for classification and one neuron for regression, to the final embeddings of $x_t$ (i.e., $\{e_{ij}\}$ for edges and $\{h_i\}$ for nodes) for discrete and continuous diffusion models, respectively.

## C   Experiment Details

### C.1   Training Details for Supervised Learning

Since we leverage the trained checkpoints introduced by DIFUSCO (Sun & Yang, 2023) and T2T (Li et al., 2023), we adopt the datasets and training procedures mentioned in DIFUSCO as shown in Table 8. This approach ensures consistency with previous work and provides a solid foundation for our RL fine-tuning experiments.

| Training Details | TSP-50 | TSP-100 | TSP-500 | TSP-1k | TSP-10k | SATLIB | ER-[700-800] |
|---|---|---|---|---|---|---|---|
| Number of epochs | 50 | 50 | 50 | 50 | 50 | 50 | 50 |
| Number of instances | 1502000 | 1502000 | 128000 | 64000 | 6400 | 49500 | 163840 |
| Batch size | 512 | 256 | 64 | 64 | 8 | 128 | 32 |
| Learning rate schedule | Cosine schedule starting from 2e-4 and ending at 0 | | | | | | |
| Curriculum learning | No | No | Yes | Yes | Yes | No | No |
| Initialization | - | - | TSP-100 | TSP-100 | TSP-500 | - | - |

Table 8: DIFUSCO Training Details for different tasks

### C.2   Training Details for RL fine-tuning

Most hyperparameters remain consistent in all experiments, with the primary variation in the number of training epochs as shown in Table 9. For RL fine-tuning starting from FastT2T within the CADO framework, as shown in Table 10, we use a relatively small number of epochs to verify the correctness and generalization of our framework.

Table 9: RL Fine-Tuning Details for CADO for different CO tasks.

| RL Fine-Tuning Details | TSP | | | | MIS | |
|---|---|---|---|---|---|---|
| | 100 | 500 | 1k | 10k | SAT | ER |
| Number of epochs | 3000 | 6000 | 6000 | 1000 | 5000 | 2500 |
| Number of samples in each epoch | 512 | | | | | |
| Batch size | 64 | | | | | |
| Learning rate | 1e-5 | | | | | |
| Denoising step (Train) | 10 | | | | | |
| LoRA Rank | 2 | | | | | |
| Number of Selective layers | 1 | | | | | |

Table 10: RL Fine-Tuning Details for different tasks from pre-trained model FastT2T (1,1).

| RL Fine-Tuning Details | TSP | | | MIS | |
|---|---|---|---|---|---|
| | 100 | 500 | 1k | SAT | ER |
| Number of epochs | 1500 | 1500 | 1500 | 1500 | 1500 |
| Number of samples in each epoch | 512 | | | | |
| Batch size | 64 | | | | |
| Learning rate | 1e-5 | | | | |
| Denoising step (Train) | (1,1) | | | | |
| LoRA Rank | 2 | | | | |
| Number of Selective Layers | 1 | | | | |

### C.3 Training Time Comparisons between RL fine-tuning and SL Learning

We compare the training time of our RL fine-tuning approach (CADO) against the standard SL training of DIFUSCO, as presented in Table 11. For a fair comparison, our RL fine-tuning process for each task was configured to use a similar number of training samples as the original SL training. For smaller-scale TSP instances (e.g., TSP-1k and below), where the original SL dataset is relatively large, CADO achieves its superior performance in significantly less training time. For tasks with smaller datasets, such as TSP-10k and MIS, the training times are comparable. Notably, across all evaluated tasks, CADO demonstrates substantial performance gains without incurring additional, and often with reduced, computational overhead for training.

| Training Time (Hours) | TSP-100 | TSP-500 | TSP-1k | TSP-10k | SATLIB | ER-[700-800] |
|---|---|---|---|---|---|---|
| SL Learning | 307 | 414 | 318.5 | 427 | 15 | 30 |
| RL Fine-Tuning | 5 | 41 | 148 | 434 | 30 | 16 |

Table 11: DIFUSCO Training Time Comparisons Between Various CO Tasks.

## D Additional Experiments and Analysis

### D.1 Comparison with Cost-aware diffusion Solvers

We hypothesize that direct objective alignment during training is fundamentally more effective than post-hoc cost-guidance applied during inference. To test this, we conduct a rigorous comparative study between CADO and leading heatmap-based solvers (DIFUSCO, T2T/FastT2T). The central challenge in comparing these methods is to disentangle the contribution of the core learning algorithm from the powerful search heuristics often used in concert. Therefore, we establish a controlled experimental testbed where auxiliary search techniques (e.g., Local Rewrite, 2-opt) are applied identically across all models. This ensures an apples-to-apples comparison, isolating the true impact of the underlying heatmap generation strategy. Moreover, we test the versatility of our approach by fine-tuning not only our base model (DIFUSCO) but also a pre-trained FastT2T model. This serves to show that CADO is not a model-specific trick, but a general framework for resolving the objective mismatch inherent in any SL-based heatmap solver.

### D.2 Taxonomy of Search Techniques

To ensure a fair comparison, it is crucial to understand the roles of different search techniques employed by heatmap-based solvers. We categorize them based on their applicability and function:

- **Cost-guided denoising process (CS)**: This is a post-hoc inference technique specific to well-trained SL models like T2T. Similar to classifier guidance, it steers the generation process towards lower-cost solutions using a predefined energy function, $l(\boldsymbol{x}, c^\star|g)$, without any additional training:

$$\tilde{\pi}_\theta(\boldsymbol{x}|c^\star, g) \propto \tilde{\pi}_\theta(\boldsymbol{x}|g) l(\boldsymbol{x}, c^\star|g).$$

  Its primary limitation is being decoder-blind; it guides the heatmap based on a cost proxy, not the final decoded solution cost. Its effectiveness is also fundamentally constrained by the quality of the initial SL-trained model.

- **Local Rewrite (LR)**: A diffusion-specific iterative refinement method. It perturbs a solution by adding noise and then denoises it, a process that can discover better solutions. As a general technique, it can be applied to any diffusion-based solver, including DIFUSCO, T2T, and CADO.

- **2-opt**: A classic local search heuristic for the TSP. It iteratively swaps pairs of edges in a decoded tour to improve solution quality. While simple, it is highly effective when combined with a strong initial solution from a heatmap-based solver. As a problem-specific but model-agnostic heuristic, it can be applied to the output of any solver.

The key distinction for our analysis lies in their applicability. Local Rewrite and 2-opt are general-purpose search components that can be integrated into CADO and our baselines alike. In contrast, Cost-Guided Denoising is an approach fundamentally tied to the post-hoc correction paradigm of T2T while RL fine-tuning is tied to the CADO framework. Therefore, to precisely isolate the difference between T2T's post-hoc guidance and CADO's direct objective alignment, we conduct controlled experiments. In these experiments, the general search techniques (LR and 2-opt) are applied identically to both models (or disabled for both). This setup creates a direct comparison between the two core cost-awareness strategies: post-hoc Cost-Guided Search versus our end-to-end RL fine-tuning.

## D.3 Controlled Experimental Settings for Fair Comparison

Since CADO shares its GNN architecture with prior works like DIFUSCO and T2T, the quality of the generated heatmap is fundamentally tied to the effectiveness of the neural network's training. However, the final performance of diffusion-based CO solvers is not determined by the heatmap alone. It is significantly influenced by various inference-time components, including the number of denoising steps and the use of auxiliary search techniques. Specifically, general-purpose heuristics such as Local Rewrite (LR) and 2-opt can substantially impact solution quality, often confounding the contribution of the core heatmap generation model. To dissect the true effect of each component and ensure a fair, rigorous comparison, we establish a controlled experimental testbed. We systematically evaluate all baselines under various settings, including configurations where these auxiliary techniques are either applied identically to all methods or disabled entirely. This controlled approach allows us to isolate the impact of our core contribution—RL-based objective alignment—from confounding factors. The detailed inference configurations for each algorithm are specified in Table 12.

Table 12: Detailed inference configurations for diffusion-based solvers. This table outlines the specific parameters used for each method during inference. Initial Denoising Timesteps and Local Rewrite Timesteps define the core generation process. Number of Decoding refers to the total number of solution candidates generated. **Number of 2-opt (w/ 2-opt)** specifies the number of 2-opt applications in our main experiments, while **Number of 2-opt (w/o 2-opt)** is for the ablation study without the heuristic.

| Algorithms | Methodology | Initial Generation iterations | Initial Denoising Timesteps | Local Rewrite iterations | Local Rewrite Denoising Timesteps | Total Denoising Timesteps | Number of Decoding | Number of 2-opt (w/o 2-opt) | Number of 2-opt (w/ 2-opt) |
|---|---|---|---|---|---|---|---|---|---|
| DIFUSCO | SL | 1 | 50 | 0 | 0 | 50 | 1 | 0 | 1 |
| CADO-L | SL + RL | 1 | 20 | 0 | 0 | 20 | 1 | 0 | 1 |
| T2T | SL+CS+LR | 1 | 20 | 3 | 10 | 50 | 4 | 0 | 4 |
| FastT2T (5,3) | SL+CS+LR | 5 | 1 | 3 | 1 | 8 | 8 | 0 | 4 |
| CADO | SL+RL+LR | 1 | 20 | 3 | 10 | 50 | 4 | 0 | 4 |
| FastT2T (1,0) | SL+CS | 1 | 1 | 0 | 0 | 1 | 1 | 0 | 1 |
| CADO + FastT2T (1,0) | SL+RL+LR | 1 | 1 | 0 | 0 | 1 | 1 | 0 | 1 |
| FastT2T (1,1) | SL+CS+LR | 1 | 1 | 1 | 1 | 2 | 2 | 0 | 2 |
| CADO + FastT2T (1,1) | SL+RL+LR | 1 | 1 | 1 | 1 | 2 | 2 | 0 | 2 |

## D.4 Analysis for Simulation Results

Among heatmap-based CO solvers, several approaches incorporate cost information with motivations similar to CADO, including DIMES (Qiu et al., 2022), T2T (Li et al., 2023), and FastT2T (Li et al., 2024). In this section, we analyze these baselines together with the purely supervised DIFUSCO (Sun & Yang, 2023).

## D.5 Analysis of Heatmap Quality without 2-opt Heuristic

To isolate the intrinsic quality of the generated heatmaps, we first conduct a comparative analysis without the powerful search heuristic, 2-opt. This experiment directly evaluates the effectiveness of the underlying heatmap generation strategy of each model. The results, presented in Table 13, offer clear observations on the effectiveness of our approach.

Table 13: Performance comparison of algorithms across various TSP and MIS problem instances. All algorithms are evaluated without the 2-opt heuristic.

| Algorithms (w/o 2-opt) | Methodology | TSP-100 | | TSP-500 | | TSP-1k | | TSP-10k | | MIS-SAT | | MIS-ER | |
|---|---|---|---|---|---|---|---|---|---|---|---|---|---|
| | | Length ↓ | Drop ↓ | Length ↓ | Drop↓ | Length ↓ | Drop↓ | Length↓ | Drop↓ | Length↓ | Drop↓ | Length↓ | Drop↓ |
| DIFUSCO | SL | 7.84 | 1.01% | 18.11 | 9.41% | 25.72 | 11.24% | 98.15 | 36.75% | 424.56 | 0.33% | 36.55 | 18.53% |
| CADO-L | SL+RL | 7.77 | 0.19% | 17.04 | 2.96% | 24.55 | 6.17% | 78.72 | 9.83% | 425.27 | 0.16% | 42.96 | 4.25% |
| T2T | SL+CS+LR | 7.77 | 0.18% | 17.69 | 6.92% | 25.39 | 9.83% | - | - | 425.02 | 0.22% | 39.56 | 11.83% |
| FastT2T (5,3) | SL+CS+LR | 7.76 | 0.07% | 17.45 | 5.45% | 25.18 | 8.90% | - | - | - | - | 40.61 | 9.52% |
| CADO | SL+RL+LR | 7.76 | 0.06% | 16.77 | 1.35% | 24.06 | 4.05% | - | - | 425.43 | 0.12% | 43.62 | 2.78% |
| FastT2T (1,0) | SL+CS | 7.91 | 1.99% | 18.04 | 8.97% | 26.68 | 15.41% | - | - | - | - | 36.88 | 17.82% |
| CADO + FastT2T (1,0) | SL+RL+LR | 7.86 | 1.32% | 17.69 | 6.87% | 25.10 | 8.55% | - | - | - | - | 39.82 | 11.25% |
| FastT2T (1,1) | SL+CS+LR | 7.78 | 0.49% | 17.58 | 6.22% | 25.94 | 12.2% | - | - | - | - | 40.29 | 10.21% |
| CADO + FastT2T (1,1) | SL+RL+LR | 7.77 | 0.16% | 17.23 | 4.11% | 24.61 | 6.46% | - | - | - | - | 41.16 | 8.28% |

### D.5.1  CADO's RL Fine-Tuning Outperforms Baselines

Across all benchmarks, CADO demonstrates a significant performance advantage. Notably, even the ablated CADO-L—which forgoes Local Rewrite and thus requires only 40% of the computational budget of T2T—consistently outperforms both the fully-equipped T2T and the baseline DIFUSCO. This result underscores the effectiveness of our direct objective alignment.

### D.5.2  Direct Objective Alignment vs. Post-Hoc Guidance

The performance gap becomes even more pronounced in a direct comparison between CADO and T2T, which share identical settings (including Local Rewrite) and the same pre-trained model. On TSP-500/1k, CADO achieves drop of just 1.35%/4.05%. On the other hand, T2T's post-hoc cost guidance yields much larger gaps of 6.92%/9.83%. This wide disparity supports our hypothesis: directly optimizing for the true, post-decoding cost via RL fine-tuning is more effective than applying post-hoc corrections to a cost-blind heatmap.

### D.5.3  Generalization and Robustness of the CADO Framework

To validate the general applicability of our framework, we fine-tuned a different pre-trained model, FastT2T. The results confirm that CADO consistently and substantially improves upon the original models, whether Local Rewrite is used or not. This is particularly notable for the FastT2T(1,0) case, which uses a single denoising step and effectively acts as a standard regression-based heatmap-based solver. CADO's strong performance in this setting demonstrates that our RL fine-tuning is a robust and principled method that works across different model architectures and configurations, not a trick specific to one setup.

### D.6  Analysis of Heatmap Quality with 2-opt Heuristic

To assess the practical utility of our framework, we evaluate all methods in a more realistic scenario where a powerful post-hoc heuristic, 2-opt, is applied to the generated solutions. Table 14 shows that CADO consistently outperforms or remains highly competitive with all baselines, reaffirming its state-of-the-art performance. We observe that the performance gap between CADO and other methods is narrower compared to the experiments without 2-opt. This is an expected and insightful result. A strong local search heuristic like 2-opt can partially compensate for deficiencies in a less optimal initial heatmap, thus elevating the performance of all solvers. However, the final solution quality is still fundamentally anchored by the quality of the initial heatmap. CADO's superior performance, even after this normalization by 2-opt, suggests that generating a better-aligned initial solution through our cost-aware framework is an important factor for achieving strong results. This confirms that CADO is practically effective in standard CO solver pipelines.

## E  TSPLIB experiment

To further demonstrate the robustness and consistency of our method on real-world problem instances, we provide a detailed per-instance performance breakdown on the TSPLIB benchmark. Table 15 covers instances with 50–200 nodes, while Table 16 covers larger instances from 200–1000 nodes. Across this diverse set of

Table 14: Performance comparison of algorithms (w/ 2-opt) across various TSP instances. All algorithms are employed **with 2-opt**.

| Algorithms (w/ 2-opt) | Methodology | TSP-100 | | TSP-500 | | TSP-1k | | TSP-10k | |
|---|---|---|---|---|---|---|---|---|---|
| | | Length ↓ | Drop ↓ | Length ↓ | Drop↓ | Length ↓ | Drop↓ | Length↓ | Drop↓ |
| DIFUSCO | SL+2-opt | 7.78 | 0.41% | 16.81 | 1.55% | 23.55 | 1.86% | 73.99 | 3.10% |
| CADO-L | SL+RL+2-opt | **7.77** | **0.12%** | **16.71** | **0.96%** | **23.44** | **1.39%** | **73.69** | **2.68%** |
| T2T | SL+CS+LR+2-opt | 7.76 | 0.06% | 16.68 | 0.83% | 23.41 | 1.26% | - | - |
| FastT2T (5,3) | SL+CS+2-opt | 7.76 | **0.01%** | 16.66 | 0.65% | 23.35 | 0.99% | - | - |
| CADO | SL+RL+LR+2-opt | 7.76 | 0.06% | **16.65** | **0.61%** | **23.32** | **0.88%** | - | - |
| FastT2T (1,0) | SL+CS+LR + 2-opt | 7.77 | 0.18% | 16.74 | 1.15% | 26.68 | 15.4% | - | - |
| CADO + FastT2T (1,0) | SL+RL+LR +2-opt | 7.77 | 0.14% | 16.73 | 1.06% | 23.45 | 1.43% | - | - |
| FastT2T (1,1) | SL+CS+LR + 2-opt | 7.77 | 0.09% | 16.74 | 1.12% | 23.42 | 1.30% | - | - |
| CADO + FastT2T (1,1) | SL+RL+LR + 2-opt | **7.76** | **0.02%** | 16.67 | 0.70% | 23.37 | 1.08% | - | - |

| Name | Optimal Len | AM | | POMO | | Sym-NCO | | ELG | | HierTSP | | DIFUSCO | | T2T | | DEITSP | | CADO | |
|---|---|---|---|---|---|---|---|---|---|---|---|---|---|---|---|---|---|---|---|---|
| | | Length | Drop ↓ | Length | Drop ↓ | Length | Drop ↓ | Length | Drop ↓ | Length | Drop ↓ | Length | Drop ↓ | Length | Drop ↓ | Length | Drop ↓ | Length | Drop ↓ |
| berlin52 | 7542.000 | 7856.426 | 4.169 | 7544.366 | 0.031 | 7544.662 | 0.035 | 7544.365 | 0.031 | 7544.662 | 0.035 | 7544.366 | 0.031 | 7544.366 | 0.031 | 7544.366 | 0.031 | 7544.365 | 0.031 |
| bier127 | 118282.000 | 125270.101 | 5.908 | 123319.180 | 4.259 | 123993.359 | 4.829 | 122855.531 | 3.867 | 124093.656 | 4.913 | 119152.524 | 1.040 | 119050.160 | 0.649 | 119367.517 | 0.918 | 118760.281 | 0.404 |
| ch130 | 6110.000 | 6304.420 | 3.182 | 6122.857 | 0.210 | 6118.715 | 0.143 | 6157.908 | 0.784 | 6157.908 | 0.784 | 6190.983 | 1.325 | 6122.493 | 0.204 | 6126.887 | 0.276 | 6114.782 | 0.078 |
| ch150 | 6528.000 | 6827.244 | 4.584 | 6562.099 | 0.522 | 6567.454 | 0.604 | 6583.826 | 0.855 | 6578.063 | 0.767 | 6585.461 | 0.880 | 6580.099 | 0.798 | 6564.298 | 0.556 | 6555.628 | 0.423 |
| eil101 | 629.000 | 647.832 | 2.994 | 640.596 | 1.844 | 640.212 | 1.782 | 643.840 | 2.359 | 642.830 | 2.199 | 641.458 | 1.981 | 642.587 | 2.155 | 644.122 | 2.404 | 640.264 | 1.790 |
| eil51 | 426.000 | 432.935 | 1.628 | 431.953 | 1.397 | 431.953 | 1.397 | 430.746 | 1.114 | 429.484 | 0.818 | 431.271 | 1.237 | 429.484 | 0.818 | 433.943 | 1.747 | 428.981 | 0.699 |
| eil76 | 538.000 | 548.717 | 1.992 | 544.369 | 1.184 | 544.652 | 1.236 | 549.433 | 2.125 | 547.020 | 1.677 | 545.048 | 1.310 | 556.769 | 3.489 | 544.369 | 1.184 | 544.369 | 1.183 |
| kroA100 | 21282.000 | 22136.898 | 4.017 | 21396.438 | 0.538 | 21357.164 | 0.353 | 21307.422 | 0.119 | 21487.941 | 0.968 | 21285.443 | 0.016 | 21307.422 | 0.119 | 21307.422 | 0.119 | 21285.443 | 0.016 |
| kroA150 | 26524.000 | 27527.668 | 3.784 | 26734.875 | 0.795 | 26890.738 | 1.383 | 26805.207 | 1.060 | 26995.578 | 1.778 | 26578.099 | 0.204 | 26525.031 | 0.004 | 26804.307 | 1.057 | 26525.031 | 0.003 |
| kroA200 | 29368.000 | 31454.890 | 7.106 | 29984.789 | 2.100 | 30206.396 | 2.855 | 29831.221 | 1.577 | 30149.805 | 2.662 | 29583.978 | 0.735 | 30033.710 | 2.267 | 29543.848 | 0.599 | 29469.600 | 0.345 |
| kroB100 | 22141.000 | 23279.490 | 5.142 | 22275.605 | 0.608 | 22374.285 | 1.054 | 22280.641 | 0.631 | 22275.350 | 0.607 | 22533.048 | 1.771 | 22645.401 | 2.278 | 22268.685 | 0.577 | 22508.477 | 1.659 |
| kroB150 | 26130.000 | 26766.788 | 2.437 | 26635.195 | 1.933 | 26816.086 | 2.626 | 26374.766 | 0.937 | 26638.965 | 1.948 | 26284.696 | 0.592 | 26244.185 | 0.438 | 26319.820 | 0.726 | 26148.483 | 0.070 |
| kroB200 | 29437.000 | 31951.214 | 8.541 | 30428.590 | 3.369 | 30563.463 | 3.827 | 29904.586 | 1.588 | 30551.854 | 3.787 | 30204.181 | 2.606 | 29841.325 | 1.374 | 29639.831 | 0.689 | 29518.668 | 0.277 |
| kroC100 | 20749.000 | 20950.680 | 0.972 | 20832.773 | 0.404 | 20959.719 | 1.016 | 20770.041 | 0.101 | 20829.061 | 0.386 | 20773.073 | 0.116 | 20750.763 | 0.008 | 21293.943 | 2.626 | 20750.755 | 0.008 |
| kroD100 | 21294.000 | 21872.558 | 2.717 | 21719.195 | 1.997 | 21635.557 | 1.604 | 21532.164 | 1.118 | 21772.988 | 2.249 | 21320.974 | 0.127 | 21294.291 | 0.001 | 21375.452 | 0.383 | 21294.290 | 0.001 |
| kroE100 | 22068.000 | 22392.400 | 1.470 | 22380.355 | 1.415 | 22346.006 | 1.260 | 22237.137 | 0.766 | 22260.621 | 0.873 | 22372.583 | 1.380 | 22362.472 | 1.334 | 22424.825 | 1.631 | 22173.068 | 0.476 |
| lin105 | 14379.000 | 14629.051 | 1.739 | 14494.633 | 0.804 | 14648.303 | 1.873 | 14467.038 | 0.612 | 14720.170 | 2.373 | 14432.139 | 0.370 | 14382.996 | 0.028 | 14382.996 | 0.028 | 14382.995 | 0.027 |
| pr107 | 44303.000 | 46045.437 | 3.933 | 44897.246 | 1.341 | 45324.367 | 2.305 | 44960.422 | 1.484 | 44958.227 | 1.479 | 45551.263 | 2.818 | 44590.338 | 0.649 | 44519.916 | 0.490 | 44729.978 | 0.963 |
| pr124 | 59030.000 | 61200.533 | 3.677 | 59091.836 | 0.105 | 59123.277 | 0.158 | 59181.652 | 0.257 | 59520.555 | 0.831 | 59181.436 | 1.329 | 59774.850 | 1.262 | 59607.740 | 0.979 | 59162.341 | 0.224 |
| pr136 | 96772.000 | 101672.534 | 5.064 | 97485.609 | 0.737 | 97513.648 | 0.766 | 97733.406 | 0.993 | 98391.383 | 1.673 | 97664.229 | 0.922 | 96925.555 | 0.159 | 98097.238 | 1.369 | 96784.010 | 0.012 |
| pr144 | 58537.000 | 63009.812 | 7.641 | 58828.539 | 0.498 | 59043.486 | 0.865 | 58859.398 | 0.551 | 58795.152 | 0.441 | 58853.339 | 0.540 | 59287.317 | 1.282 | 58604.905 | 0.116 | 58646.905 | 0.187 |
| pr152 | 73682.000 | 79203.729 | 7.494 | 74440.711 | 1.030 | 76061.063 | 3.229 | 73720.609 | 0.052 | 74485.031 | 1.090 | 75613.977 | 2.622 | 74640.127 | 1.260 | 73683.641 | 0.002 | 74588.441 | 1.230 |
| pr76 | 108159.000 | 109041.577 | 0.816 | 108159.438 | 0.000 | 108591.000 | 0.399 | 108444.047 | 0.264 | 108428.906 | 0.250 | 110019.494 | 1.720 | 109707.304 | 1.490 | 109086.647 | 0.926 | 109081.840 | 0.853 |
| rat195 | 2323.000 | 2483.124 | 6.893 | 2558.470 | 10.136 | 2569.150 | 10.596 | 2400.750 | 3.348 | 2492.917 | 7.315 | 2392.573 | 2.995 | 2365.187 | 1.816 | 2355.277 | 1.196 | 2346.410 | 1.007 |
| rat99 | 1211.000 | 1243.031 | 2.645 | 1273.635 | 5.172 | 1264.701 | 4.434 | 1248.142 | 3.067 | 1240.427 | 2.430 | 1220.098 | 0.751 | 1219.244 | 0.681 | 1219.244 | 0.681 | 1219.243 | 0.680 |
| st70 | 675.000 | 686.725 | 1.737 | 677.110 | 0.313 | 677.110 | 0.313 | 677.110 | 0.313 | 677.110 | 0.313 | 677.642 | 0.391 | 677.194 | 0.325 | 677.110 | 0.313 | 677.109 | 0.312 |
| Mean | 31466.115 | 32703.968 | 3.934 | 31902.325 | 1.386 | 32069.482 | 1.917 | 31825.516 | 1.142 | 32025.602 | 1.778 | 31870.251 | 1.284 | 31750.494 | 0.903 | 31777.827 | 0.781 | **31610.837** | **0.460** |

Table 15: Comparison on TSPLIB instances with 50–200 nodes.

| Name | Optimal Len | DIFUSCO | | T2T | | CADO | |
|---|---|---|---|---|---|---|---|
| | | Length | Drop ↓ | Length | Drop ↓ | Length | Drop ↓ |
| a280 | 2579.000 | 2714.408 | 5.25 | 2676.005 | 3.761 | 2589.557 | 0.409 |
| d493 | 35002.000 | 35999.523 | 2.85 | 35831.093 | 2.369 | 35646.508 | 1.841 |
| d657 | 48912.000 | 50127.103 | 2.484 | 50095.534 | 2.42 | 49693.430 | 1.598 |
| fl417 | 11861.000 | 12284.844 | 3.573 | 12172.422 | 2.626 | 12082.369 | 1.866 |
| p654 | 34643.000 | 35167.664 | 1.514 | 35011.017 | 1.062 | 34982.382 | 0.980 |
| lin318 | 42029.000 | 43202.828 | 2.793 | 42465.811 | 1.039 | 42654.013 | 1.487 |
| pr1002 | 259045.000 | 269491.268 | 4.033 | 266756.447 | 2.977 | 264527.769 | 2.117 |
| pcb442 | 50778.000 | 52267.383 | 2.933 | 51418.603 | 1.262 | 51135.646 | 0.704 |
| pr226 | 80369.000 | 81498.530 | 1.405 | 80863.979 | 0.616 | 80678.390 | 0.385 |
| pr264 | 49135.000 | 49876.086 | 1.508 | 49648.489 | 1.045 | 49184.913 | 0.102 |
| pr439 | 107217.000 | 111463.546 | 3.961 | 109562.481 | 2.188 | 108219.416 | 0.935 |
| pr299 | 48191.000 | 49648.887 | 3.025 | 48806.939 | 1.278 | 48615.520 | 0.881 |
| rat575 | 6773.000 | 6946.187 | 2.557 | 6893.903 | 1.785 | 6860.361 | 1.290 |
| rat783 | 8806.000 | 9035.575 | 2.607 | 8986.024 | 2.044 | 8973.545 | 1.903 |
| rd400 | 15281.000 | 15569.444 | 1.888 | 15434.598 | 1.005 | 15314.435 | 0.219 |
| tsp225 | 3916.000 | 3988.991 | 1.864 | 3931.472 | 0.395 | 3923.511 | 0.192 |
| ts225 | 126643.000 | 129143.181 | 1.974 | 128512.863 | 1.476 | 126725.437 | 0.065 |
| u574 | 36905.000 | 37557.611 | 1.768 | 37414.204 | 1.38 | 37283.200 | 1.025 |
| u724 | 41910.000 | 43019.140 | 2.646 | 42656.670 | 1.782 | 42504.571 | 1.419 |
| Mean | 53157.632 | 54684.326 | 2.665 | 54165.187 | 1.711 | **53768.157** | **1.149** |

Table 16: Comparison on TSPLIB instances with 200–1000 nodes.

instances, CADO consistently achieves the lowest drop, outperforming all diffusion-based baselines. This confirms that CADO's superiority is not an artifact of averaging over a specific data distribution but a consistent advantage across various problem structures and scales.

# F    Learning Curves of CADO across CO Tasks

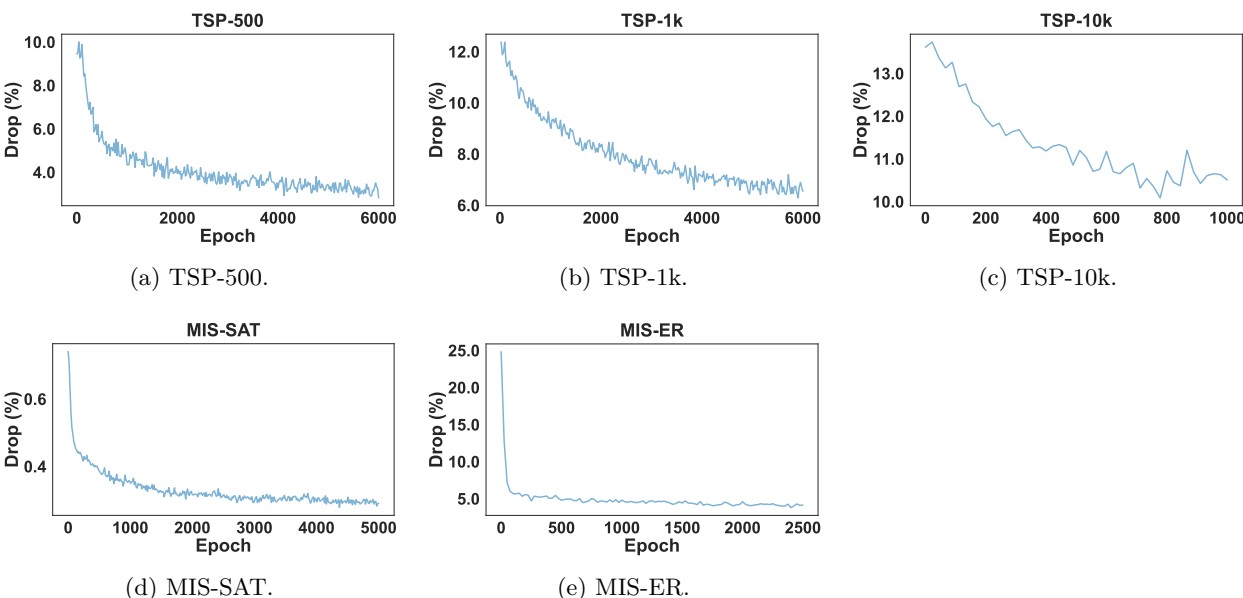

(a) TSP-500.                    (b) TSP-1k.                    (c) TSP-10k.

(d) MIS-SAT.                    (e) MIS-ER.

Figure 5: Learning curves of CADO-L across different CO tasks, measured in Drop (%). Each curve is from a single representative seed; the main text (Figure 3) reports the average of 4 independent runs.

Figure 5 presents the training curves of CADO-L across all five tasks, recorded from the actual checkpoint during RL fine-tuning with a single fixed seed. Across all tasks, the Drop consistently decreases throughout training, demonstrating that CADO's RL fine-tuning stably and reliably improves solution quality regardless of the problem type or scale.

