# OpenReview forum: "CADO: From Imitation to Cost Minimization for Heatmap-based Solvers in Combinatorial Optimization"
_TMLR — Accepted by TMLR_

### Review · Reviewer_k2u6 · 2026-03-03

**Summary Of Contributions:**

The authors propose to use reinforcement learning to train neural heatmap generators for combinatorial optimization.

Recently it was proposed to solve combinatorial optimization problems with heatmaps generated by image-generating neural networks (followed by algorithmic post-processing to generate an actual solution). (Presumably this applies to problems for which the solution can be neatly represented by a matrix). However, this usually relies on supervised learning based on a dataset of solved instances.

The authors point out that supervised learning introduces difficulties due to the difficulty of backpropagating over the post-processing part and the resulting approximations (IIUC it also requires a dataset of existing solved instances, which apparently is not strictly the case for the method they propose).

They suggest to use reinforcement learning (RL) instead, namely the REINFORCE algorithm, which can make use of the final post-processing output's score directly.

(There seems to be an assumption that we have access to a perfect scoring method that estimates the cost/validity of any proposed solution at negligible cost.)

RL eliminates the approximations of the post-processing state, and also does not strictly need a dataset of existing solved instances (as long as a verifier for any proposed solution is available). However the authors also show that RL can use such a dataset, by using the cost of the optimal solutions for each instance as a baseline for REINFORCE (it is known that REINFORCE benefits from accurate baselines).

The authors produce numerous experiments and ablations to analyze their method. They show that the method produces superior results to supervised learning alone, that it still requires supervised pre-training for best performance, that including baselines from a dataset of optimal solutions is beneficial, etc.

**Audience:**

Yes

**Audience Explanation:**

Neural approaches to combinatorial optimization are of great interest, although I am not an expert in this area.

**Broader Impact Concerns:**

No broader impact concerns I can think of.

**Claims And Evidence:**

Yes

**Claims Explanation:**

The experiments are numerous and address the questions I was having while reading the paper. Assuming the data was correctly collected, they seem to support the author's claims.

**Requested Changes:**

Given my low level of expertise in this area, I have very few changes to suggest.

- Make it clear which kind of problems are amenable to heatmap-based approaches (I suspect that it implies that the solutions be representable as matrices, but I may be wrong)

- The paper seems to assume the availability  of a perfect scoring method that estimates the cost/validity of any proposed solution at negligible cost. This is not a very onerous assumption (NP problems, by definition, have solutions verifiable in polynomial time) but it should be made explicit.

- In appendix B, I suppose capital A in the third paragraph is the lower-case alpha of equation 19?

- Unless I missed it, it is not explicitly stated how the supervised pre-training occurs. From various hints in the text, it seems that the pre-trained network is simply a publicly available DIFUSCO checkpoint. This should be made explicit.

- "GNN" is undefined in section 4.3 (I'm assuming it means "Graph Neural Network")

---

> ### Author Response · Authors · 2026-04-04
>
> We are grateful to the reviewer for the careful and insightful review. All modifications in the revised manuscript are highlighted in blue for easy identification.
>
> ## Q1. On the scope of heatmap-based approaches
>
> *"Make it clear which kind of problems are amenable to heatmap-based approaches (I suspect that it implies that the solutions be representable as matrices, but I may be wrong)"*
>
> ## [Response]
>
> As defined in Section 2.1, the mathematical representation of the solution x ∈ X_g depends on the nature of the specific combinatorial optimization problem:
>
> - For edge-centric problems like the TSP, the solution is indeed represented as an n × n adjacency matrix.
> - Conversely, for node-centric problems like the MIS, the solution is represented as a 1D binary vector of size |V_g|, where each element indicates the inclusion of a specific node in the independent set.
>
> Therefore, in principle, heatmap-based approaches are broadly amenable to any problem where the discrete solution space can be formulated as a fixed-dimensional probability distribution over the graph's components (either nodes or edges). We have made this scope explicitly clear in the revised Section 2.1.
>
> ## Q2. On the assumption of a perfect scoring method
>
> *"The paper seems to assume the availability of a perfect scoring method that estimates the cost/validity of any proposed solution at negligible cost. This is not a very onerous assumption (NP problems, by definition, have solutions verifiable in polynomial time) but it should be made explicit."*
>
> ## [Response]
>
> As the reviewer correctly notes, the CO problems we address (TSP, MIS) have cost functions that can be evaluated exactly in polynomial time, making reward computation straightforward. We note that this assumption is shared by the vast majority of CO problems commonly studied in the NCO literature [1,2,3,4] (e.g., CVRP, Knapsack, OVRP), where solution cost evaluation is straightforward. Nonetheless, we have made this assumption explicit in the revised manuscript for clarity.
>
> ## Q3. On minor issues
>
> *"(1) Unless I missed it, it is not explicitly stated how the supervised pre-training occurs. (2) In appendix B, I suppose capital A in the third paragraph is the lower-case alpha of equation 19? (3) 'GNN' is undefined in section 4.3."*
>
> ## [Response]
>
> All three points have been corrected in the revised manuscript: (1) We now explicitly state in Section 4 that our SL initialization uses the publicly available DIFUSCO checkpoints. (2) The capital 'A' typo in Appendix B has been corrected to the lower-case α of Equation 19. (3) "GNN" (Graph Neural Network) is now defined at its first appearance in Section 4.3.
>
> **References:**
> [1] Kool et al., "Attention, Learn to Solve Routing Problems!" ICLR 2019
> [2] Sun & Yang, "DIFUSCO: Graph-based Diffusion Solvers for Combinatorial Optimization," NeurIPS 2023
> [3] Qiu et al., "DIMES: A Differentiable Meta Solver for Combinatorial Optimization Problems," NeurIPS 2022
> [4] Kwon et al., "POMO: Policy Optimization with Multiple Optima for Reinforcement Learning," NeurIPS 2020

---

### Review · Reviewer_v9jt · 2026-03-10

**Summary Of Contributions:**

The paper focuses on learning to solve combinatorial optimization and argues for a challenge in existing family of "heatmap" methods. The challenge argued is that there is a drop in solution quality due to gap in continuous relaxation output space of neural networks and the discrete decoded final output. The proposed approach is a straightforward application of RL based finetuning of diffusion models to be more decoder aware.

**Audience:**

No

**Audience Explanation:**

I think the paper requires a lot of work to be useful to TMLR audience.

**Claims And Evidence:**

No

**Claims Explanation:**

- The empirical evidence provided in the paper in section 3 is not convincing. First, the hypotheses which form the premise of the figures itself are a bit weak since they rely on hamming distance which is not all the right metric for such combinatorial optimization problems. Hamming distance ignores geometry and constraints and the problems would be much simpler if these statements were true. So the near-zero correlation between Hamming distance and cost may partly be a property of the chosen metric, which is not a diagnosis of some underlying issue with the "supervised learning" approach. This makes it hard to be convinced about the main premise of the paper of "objective mismatch".

- The RL based finetuning is straightforward and conceptually direct. The MDP is quite common in existing RL plus diffusion papers and not 'novel' as the paper claims.

- There are many statements made in the paper which are not justified properly or has logical issues in the justification.
  - For example, the paper mentions that "discussed in section 2.2... Unfortunately, the polynomial-time forward pass of neural networks fundamentally cannot achieve such exact imitation, owing to the NP-hardness of CO ...".  However, this is a strawman argument since the original objective as defined in the paper is to only find "near-optimal" solutions. Using NP-hardness as argument for an objective which explicitly asks for "near-optimal" solution is not justified.
  - Similar, consider interpretation of Figure 3. The paper says that because RL fine-tuning lowers drop while SL loss rises, this “decisively validates” that imitating optimal solutions is ineffective. This is much stronger statement than what the figure actually shows. Once you optimize a different objective, it is completely expected that the original surrogate loss may worsen.
  - Both these main statements about existing work are not justified or supported with prior reference: " ... it often suffers from training instability, particularly in large-scale CO problems involving high-dimensional heatmaps" and "while the SL objective L(θ) serves as a surrogate, it has been widely regarded as theoretically sound"

**Requested Changes:**

I would really encourage to consider removing certain non-scholarly phrases which are widely used in the paper like "decisively validates", "essential for unlocking", "pivotal shift", "precise objective alignment" etc.

---

> ### Author Response · Authors · 2026-04-04
>
> We sincerely thank the reviewer for the detailed and rigorous feedback. All modifications in the revised manuscript are highlighted in blue for easy identification.
>
> ## W1. On the NP-hardness argument
>
> *"The paper mentions that 'discussed in section 2.2... Unfortunately, the polynomial-time forward pass of neural networks fundamentally cannot achieve such exact imitation, owing to the NP-hardness of CO ...'. However, this is a strawman argument since the original objective as defined in the paper is to only find 'near-optimal' solutions."*
>
> ## [Response]
>
> We agree with the reviewer that the NP-hardness argument was not well-justified for our claim. In the revised manuscript, we have removed this theoretical argument and restructured Section 3 so that the two hypotheses H1 and H2 serve directly as the empirical starting point. The SL paradigm has been widely adopted under the implicit assumption that heatmaps closely approximating optimal solutions would naturally yield low-cost decoded solutions. Rather than arguing theoretical impossibility, we now focus on showing empirically that this hypothesis is not well-supported in practice (Figure 1).
>
> ## W2. On the use of Hamming distance
>
> *"The hypotheses which form the premise of the figures itself are a bit weak since they rely on hamming distance which is not all the right metric for such combinatorial optimization problems. Hamming distance ignores geometry and constraints and the problems would be much simpler if these statements were true."*
>
> ## [Response]
>
> We acknowledge this critique. These clarifications have been reflected in the revised Section 3.
>
> Our choice of Hamming distance is principled and directly motivated by the SL training paradigm we analyze. Specifically, the state-of-the-art SL-based heatmap solvers in the literature [1-5] employ element-wise cross-entropy as their SL loss. Since cross-entropy operates between continuous heatmaps and discrete labels, it cannot directly compare two discrete solutions. Hamming distance is its natural element-wise counterpart in the discrete domain.
>
> We agree with the reviewer that Hamming distance ignores geometry and constraints. However, we note that the element-wise cross-entropy loss used by SL training itself shares exactly the same limitation. To further support this, we have added Figure 1(c) in the revised manuscript, which directly plots the actual SL loss against the decoded solution drop, revealing a weak correlation (R² = 0.038). This suggests that the objective mismatch in Figure 1(b) is not solely due to the choice of metric.
>
> ## W3. On the interpretation of Figure 3
>
> *"The paper says that because RL fine-tuning lowers drop while SL loss rises, this 'decisively validates' that imitating optimal solutions is ineffective. This is much stronger statement than what the figure actually shows."*
>
> ## [Response]
> We acknowledge the reviewer's point that some divergence in the surrogate loss is expected when optimizing a different objective, and agree that "decisively validates" was too strong. We have toned down this expression in the revision.
>
> However, we note that the SL loss in Figure 3 does not merely increase temporarily. It rises monotonically (0.87 → 1.4) throughout the entire training process while solution quality steadily improves (0.6% → 0.25% drop). If the two objectives shared the same or similar optimum, we would expect the SL loss to stabilize or decrease as the model converges under the RL objective. This sustained divergence, combined with the near-zero correlation between SL loss and solution cost (R² = 0.038, Figure 1(c)) and the negligible gains from extended SL training (DIFUSCO+SFT, Table 4), suggests that the solution quality improvement achieved by RL fine-tuning is unlikely attainable through SL loss minimization alone.
>
>
> ## W4. On the straightforwardness of RL-based finetuning
>
> *"The RL based finetuning is straightforward and conceptually direct."*
>
> ## [Response]
>
> While the core RL fine-tuning formulation is indeed straightforward, CADO introduces two practical techniques that address practical challenges in applying RL fine-tuning to heatmap solvers:
>
> **Reward Strategies (SR and LCR):** CADO supports two reward strategies: the conventional Standard Reward (SR) and our Label-Centered Reward (LCR), which repurposes ground-truth costs as instance-specific REINFORCE baselines. LCR is theoretically unbiased regardless of label optimality and empirically effective even with suboptimal labels (Table 7).
>
> **Hybrid Fine-Tuning:** Combines LoRA for backbone layers with full fine-tuning for output layers. No single strategy alone is sufficient: full fine-tuning collapses, LoRA-only converges slowly, and selective-only plateaus early (Figure 4).

---

> ### Author Response · Authors · 2026-04-04
>
> ## W5. On unjustified statements about existing work
>
> *"Both these main statements about existing work are not justified or supported with prior reference: '... it often suffers from training instability, particularly in large-scale CO problems involving high-dimensional heatmaps' and 'while the SL objective L(θ) serves as a surrogate, it has been widely regarded as theoretically sound'"*
>
> ## [Response]
>
> We agree with the reviewer that both statements were imprecise. We have revised them accordingly.
>
> **"training instability in large-scale CO problems" → "lower performance across overall CO problems."** Existing RL-based heatmap solvers such as DIMES [6] and RL-Scratch consistently underperform SL-based counterparts [2,3,4,5] across all benchmarks (Tables 4, 13), which illustrates the limitation of RL in this domain. Corresponding references have been added in the revised Section 2.2.
>
> **"widely regarded as theoretically sound" → "widely adopted due to its stronger empirical performance relative to RL alternatives."** The SL objective is a surrogate loss without formal theoretical guarantees, and the original wording was an overclaim. The revised phrasing reflects the empirical, rather than theoretical, basis for the SL paradigm's dominance. This revision has been reflected in Section 2.2.
>
> ## W6. On interest to the TMLR audience
>
> *"Would at least some individuals in TMLR's audience be interested in knowing the findings of this paper?: No. I think the paper requires a lot of work to be useful to TMLR audience."*
>
> ## [Response]
>
> The objective mismatch inherent in the SL paradigm [7] has either been overlooked [1, 2, 5] or only partially addressed at inference time [3, 4] by existing heatmap solvers. Our experiments (Tables 2–7, Figures 1, 3–4) consistently show that training-time objective alignment outperforms both pure SL-based solvers and inference-time cost-guided approaches. Moreover, CADO is model-agnostic and can be applied to a wide range of SL-based heatmap solver without architectural changes (Tables 13–14). We believe these findings offer a valuable insight for the CO community. We hope that the revisions addressing the reviewer's concerns on writing quality and logical rigor will also improve the overall impression of the paper's contribution.
>
> ## W7. On non-scholarly phrases and MDP novelty claim
>
> *"I would really encourage to consider removing certain non-scholarly phrases which are widely used in the paper like 'decisively validates', 'essential for unlocking', 'pivotal shift', 'precise objective alignment' etc. The MDP is quite common in existing RL plus diffusion papers and not 'novel' as the paper claims."*
>
> ## [Response]
>
> We agree with the reviewer's suggestion. We have removed or toned down all non-scholarly phrases throughout the manuscript, including "decisively validates," "essential for unlocking," "pivotal shift," and "precise objective alignment." We have also removed the novelty claim regarding the MDP formulation, as we acknowledge that formulating diffusion denoising as an MDP is common in the RL + diffusion literature. These revisions are reflected across the revised manuscript.
>
> **References:**
> [1] Joshi et al., "An efficient graph convolutional network technique for the travelling salesman problem," arXiv preprint, 2019 (GCN)
> [2] Sun & Yang, "DIFUSCO: Graph-based diffusion solvers for combinatorial optimization," NeurIPS 2023 (DIFUSCO)
> [3] Li et al., "From distribution learning in training to gradient search in testing for combinatorial optimization," NeurIPS 2023 (T2T)
> [4] Li et al., "Fast T2T: Optimization consistency speeds up diffusion-based training-to-testing solving," NeurIPS 2024 (FastT2T)
> [5] Wang et al., "An efficient diffusion-based non-autoregressive solver for traveling salesman problem," KDD 2025 (DEITSP)
> [6] Qiu et al., "DIMES: A differentiable meta solver for combinatorial optimization problems," NeurIPS 2022
> [7] Xia et al., "Position: Rethinking post-hoc search-based neural approaches for solving large-scale traveling salesman problems," ICML 2024

---

### Review · Reviewer_T8Li · 2026-03-23

**Summary Of Contributions:**

This paper proposes CADO, a training framework for heatmap-based neural solvers for combinatorial optimization. Instead of relying purely on imitation learning from heuristic solvers, the method reformulates the training objective as cost minimization, allowing the model to optimize directly with respect to the objective of the combinatorial problem. The approach bridges imitation learning and cost-based optimization by using the heatmap representation of candidate solutions and introducing a learning objective that better aligns the model with the final optimization cost. Experiments on several combinatorial optimization tasks show that the proposed method can improve solution quality compared to standard imitation-based training.

Key Strengths：
1. Introduces a conceptually simple framework that shifts training from pure imitation toward direct cost minimization, which is more aligned with the true objective of combinatorial optimization.
2. The method is general and compatible with heatmap-based neural solvers, making it potentially applicable to multiple problems.
3. Empirical results suggest improvements over standard imitation learning baselines.

Key Weaknesses
1. The novelty appears somewhat limited, as the approach mainly reframes existing imitation-based training with a modified objective.
2. Experimental evaluation is relatively limited, making it unclear how well the method generalizes to larger-scale problems or stronger solvers.
3. Additional qualitative or diagnostic analysis would help clarify when and why the method provides improvements.

**Audience:**

Yes

**Audience Explanation:**

The paper targets a critical bottleneck in Neural Combinatorial Optimization (NCO) by pivoting from Imitation Learning to direct Cost Minimization. This shift is of high interest to some of the research community as it addresses the limitation of neural solvers being capped by the performance of their "teacher" heuristics.

1. Key Interest: Replacing imitation with objective-aligned optimization is a fundamental methodological improvement.
2. Trend Alignment: The use of diffusion-style iterative refinement for heatmaps connects generative modeling trends with structured decision-making.
3. Practical Utility: The approach offers a clear blueprint for improving hybrid solvers where neural heatmaps guide classical search algorithms.

While the experimental scale could be broader, the conceptual shift toward cost-aware training is a relevant and valuable contribution to the field.

**Broader Impact Concerns:**

I do not have ethical concerns that would require a substantial Broader Impact discussion beyond a brief statement.

**Claims And Evidence:**

No

**Claims Explanation:**

While the authors provide empirical results showing improvements over imitation learning baselines, the evidence is not fully convincing. The evaluation is limited to a relatively narrow set of experiments, making it unclear whether the proposed method generalizes to larger-scale settings or stronger baselines. In addition, the analysis is somewhat limited, and the paper does not provide sufficient evidence to clearly demonstrate the broader effectiveness of the approach. As a result, the claims are only partially supported by the current experimental results.

**Requested Changes:**

Critical:
1. Better justify novelty relative to prior objective-aware / RL-based training methods, since the current contribution may otherwise appear mainly as a modified training objective on top of an existing framework.
2. Expand experiments to larger-scale settings, stronger solvers, and more challenging benchmarks to support claims of generality.
3. Provide ablations that isolate which part of the method drives the observed gains.

Non-critical:
1. Add qualitative or diagnostic analysis to explain when and why the method outperforms imitation learning.
2. Clarify expected generalization, limitations, and scaling behavior.
3. Report compute overhead and training stability more explicitly.

---

> ### Author Response · Authors · 2026-04-04
>
> We thank the reviewer for the thoughtful and constructive feedback. All modifications in the revised manuscript are highlighted in blue for easy identification.
>
> ## W1. Novelty
>
> *"The novelty appears somewhat limited, as the approach mainly reframes existing imitation-based training with a modified objective. Better justify novelty relative to prior objective-aware / RL-based training methods."*
>
> ## [Response]
>
> We appreciate the concern and clarify our contribution below.
>
> While the core RL fine-tuning formulation is indeed straightforward, CADO introduces two practical techniques that go beyond standard RL fine-tuning:
>
> **Reward Strategies (SR and LCR):** CADO supports two reward strategies: the conventional Standard Reward (SR) and our Label-Centered Reward (LCR), which repurposes ground-truth costs as instance-specific REINFORCE baselines. LCR is theoretically unbiased regardless of label optimality and empirically effective even with suboptimal labels (Table 7).
>
> **Hybrid Fine-Tuning:** Combines LoRA for backbone layers with full fine-tuning for output layers. No single strategy alone is sufficient: full fine-tuning collapses, LoRA-only converges slowly, and selective-only plateaus early (Figure 4).
>
> Beyond RL fine-tuning itself, we believe our work makes two broader contributions to the heatmap-based CO solver literature from the perspective of directly addressing the objective mismatch:
>
>
> **Diagnostic contributions.** Prior works noted the SL–RL objective discrepancy only in an analytical manner. Our experiments (H1/H2 in Figure 1, SL loss vs. Drop in Figure 3, decoder-specific evaluation in Table 1) empirically characterize this mismatch and show that it leads to performance degradation.
>
>
> **Training-time objective alignment.** Existing SL-based heatmap solvers have largely relied on imitation of optimal labels and focused on inference-time cost integration techniques (e.g., cost-guided search in T2T and FastT2T). In contrast, CADO is, to our knowledge, the first work to directly perform objective alignment at the training stage for SL-based heatmap solvers, demonstrating that this approach is fundamentally more effective than post-hoc corrections (Table 4).
>
> ---
>
> ## W2. Experimental Scope
>
> *"Experimental evaluation is relatively limited, making it unclear how well the method generalizes to larger-scale problems or stronger solvers. Expand experiments to larger-scale settings, stronger solvers, and more challenging benchmarks to support claims of generality."*
>
> ## [Response]
>
> **Larger-scale problems.** Since our goal is to overcome the fundamental limitations of heatmap CO solvers, we evaluate on standard benchmarks widely adopted for fair comparison established in this literature [1-6], up to and including the largest scales these solvers have been tested on. For TSP, this spans 100 to 10,000 nodes; for MIS, graphs with 700–1,200+ nodes (Tables 2–3). We further validate our method on TSPLIB up to 1,000 nodes (Tables 15–16), confirming generalizability to real-world instances.
>
> **Stronger solvers.** We compare against all publicly available state-of-the-art heatmap-based solvers [1-6]. Beyond these, our evaluation includes recent autoregressive methods [7-9] and divide-and-conquer methods [10, 11] in Table 2, against which CADO remains competitive or superior even on large-scale instances. We also apply CADO to another heatmap solver, FastT2T, and observe consistent improvements (Tables 13–14).
>
> **Challenging settings.** On MIS-ER, CADO reduces the drop from 18.53% (DIFUSCO) to 2.78%. Under low-quality training data (Table 7), CADO remains robust at 1.71% while DIFUSCO degrades to 11.84%. These results indicate that CADO is particularly effective in challenging scenarios where existing heatmap solvers struggle most.
>
> **Generalization and scaling behavior.** Our experiments span two widely adopted problem types in the field (TSP, MIS), diverse graph distributions, scales from 100 to 10k, robustness under suboptimal data, and real-world instances. CADO consistently improves upon SL-based solvers across all settings. That said, like other NCO solvers including heatmap-based methods, training efficiency degrades as problem scale increases. For scales well beyond 10k nodes (e.g., TSP-100k), combining CADO with divide-and-conquer methods [10,11] would be a natural extension worth exploring, as CADO could potentially be applied to the sub-problem solvers within such pipelines.
>
> We would also be happy to conduct additional experiments on specific benchmarks or solvers the reviewer considers important, and will incorporate them during the revision period.

---

> ### Author Response · Authors · 2026-04-04
>
> ## W3. Diagnostic Analysis
>
> *"Additional qualitative or diagnostic analysis would help clarify when and why the method provides improvements. The paper does not provide sufficient evidence to clearly demonstrate the broader effectiveness of the approach."*
>
> ## [Response]
>
> **SL-loss vs Solution Cost.** We have added Figure 1(c) in the revised manuscript, which directly plots the continuous SL loss against the decoded solution cost. The correlation is even lower than those reported in Figures 1(a) and 1(b) (R² = 0.038), confirming that the objective mismatch is intrinsic to the SL paradigm.
>
> **Cost-Blindness, amplified by low-quality labels.** SL performance is tightly coupled to label quality. With suboptimal labels (1.36% known gap), DIFUSCO degrades to 11.84% and T2T recovers only to 6.99%. CADO remains robust at 1.71% (LCR). LCR outperforms SR even with noisy labels, consistent with our theoretical analysis in Section 4.2.
>
> **Decoder-Blindness, amplified by structurally complex problems.** MIS-ER graphs have high edge density, making solution quality highly sensitive to the greedy decoder's sequential choices. Since SL is oblivious to this decoder behavior, DIFUSCO achieves only 18.53% drop. CADO learns decoder-aware heatmaps, reducing the gap to 2.78%.
>
> ---
>
> ## W4. Ablations
>
> *"Provide ablations that isolate which part of the method drives the observed gains."*
>
> ## [Response]
>
> Our paper includes ablations isolating each component.
>
> **(1) RL fine-tuning vs. additional SL training (Table 4).** DIFUSCO+SFT (50% more SL epochs) yields negligible gains (11.2% → 10.1% on TSP-500), while CADO-L achieves 3.0%. The SL plateau stems from the objective mismatch, not insufficient training.
>
> **(2) Necessity of SL pre-training (Table 4).** RL-Scratch (no SL initialization) performs far worse than SL methods (18.0% vs. 11.2% on TSP-500), confirming that SL pre-training and RL fine-tuning are complementary.
>
> **(3) Standard Reward vs. Label-Centered Reward (Table 5).** Both SR and LCR substantially outperform DIFUSCO, confirming RL-based alignment as the primary driver. LCR provides an additional 10.0%–18.8% improvement over SR.
>
> **(4) Fine-tuning strategies (Figure 4).** Full-FT is unstable, LoRA-FT converges slowly, Selective-FT plateaus early. Only Hybrid-FT achieves both fast convergence and the best final performance.
>
> Collectively, these ablations show that each component makes a distinct and necessary contribution.
>
> ---
>
> ## W5. Compute Overhead
>
> *"Report compute overhead and training stability more explicitly."*
>
> ## [Response]
>
> **Training overhead.** As detailed in Table 11 of the Appendix, RL fine-tuning time is generally shorter than or comparable to the original SL pre-training time, depending on the task and dataset size.
>
> **Inference overhead.** CADO preserves the original architecture with only lightweight LoRA parameters, so inference cost is negligible. CADO-L achieves superior performance to DIFUSCO at only 40% of its inference budget (20 vs. 50 denoising steps, Table 12).
>
> **Training stability.** Hybrid-FT ensures stable training across all tasks. We provide the training curves (CADO-L) for TSP-500/1k/10k, MIS-SAT, and MIS-ER in the newly added Appendix F, all confirming stable and steady improvement.
>
> **Limitations.** CADO requires an SL-pretrained solver as its starting point, inheriting the need for a labeled dataset and the computational cost of SL pre-training in addition to RL fine-tuning. Moreover, although CADO consistently improves SL-based heatmap solvers across all evaluated tasks, its final performance is influenced by the quality of the pre-trained model, as shown in the RL-Scratch results (Table 4). We have made these limitations explicit in Section 8 of the revised manuscript.

---

> ### Author Response · Authors · 2026-04-04
>
> **References:**
> [1] Qiu et al., "DIMES: A Differentiable Meta Solver for Combinatorial Optimization Problems," NeurIPS 2022.
> [2] Sun & Yang, "DIFUSCO: Graph-Based Diffusion Solvers for Combinatorial Optimization," NeurIPS 2023.
> [3] Li et al., "From Distribution Learning in Training to Gradient Search in Testing for Combinatorial Optimization," NeurIPS 2023.
> [4] Li et al., "Fast T2T: Optimization Consistency Speeds Up Diffusion-Based Training-to-Testing Solving for Combinatorial Optimization," NeurIPS 2024.
> [5] Wang et al., "An Efficient Diffusion-Based Non-Autoregressive Solver for Traveling Salesman Problem," KDD 2025.
> [6] Li et al., "Destroy and Repair Using Hyper-Graphs for Routing," AAAI 2025.
> [7] Verdú et al., "Scaling Combinatorial Optimization Neural Improvement Heuristics with Online Search and Adaptation," AAAI 2025.
> [8] Chalumeau et al., "Combinatorial Optimization with Policy Adaptation Using Latent Space Search," NeurIPS 2023.
> [9] Chalumeau et al., "Memory-Enhanced Neural Solvers for Routing Problems," NeurIPS 2025.
> [10] Ye et al., "GLOP: Learning Global Partition and Local Construction for Solving Large-Scale Routing Problems in Real-Time," AAAI 2024.
> [11] Zheng et al., "UDC: A Unified Neural Divide-and-Conquer Framework for Large-Scale Combinatorial Optimization Problems," NeurIPS 2024.

---

### Decision · Action_Editor_e2gW · 2026-06-09

**Recommendation:** Accept as is

**Audience:**

Yes

**Audience Explanation:**

This paper will be relevant to researchers and practitioners working within the domains of neural combinatorial optimization, generative modeling for structured decision-making, and hybrid neural-classical solvers.

**Claims And Evidence:**

Yes

**Claims Explanation:**

This paper identifies an objective mismatch in supervised learning for heatmap-based combinatorial optimization solvers, which the authors categorize into "Decoder-Blindness" and "Cost-Blindness".  To resolve these limitations, the authors introduce CADO (Cost-Aware Diffusion models for Optimization), which formulates the diffusion denoising process as a Markov Decision Process to directly optimize post-decoded solution costs via reinforcement learning fine-tuning.  The proposed approach incorporates a Label-Centered Reward strategy using ground-truth costs as unbiased baselines alongside a parameter-efficient Hybrid Fine-Tuning method, achieving state-of-the-art performance across diverse optimization benchmarks.

Reviewers initially raised concerns regarding the conceptual novelty of the reinforcement learning formulation, the choice of metrics in the diagnostic analysis, and the overall scale of the experiments.  The authors successfully resolved these concerns during the rebuttal by toning down theoretical claims, adding a direct surrogate-to-cost correlation analysis, and extending evaluations to larger-scale problems and real-world datasets like TSPLIB.  One reviewer noted, and I concur, that getting reinforcement learning to work successfully and stably on a complex optimization problem is not from trivial, rendering this a valuable and useful contribution to the field.